# RNF167 activates mTORC1 and promotes tumorigenesis by targeting CASTOR1 for ubiquitination and degradation

Tingting Li[1], Xian Wang[1], Enguo Ju [1], Suzane Ramos da Silva[1], Luping Chen[1], Xinquan Zhang[1], Shan Wei[1] & Shou-Jiang Gao [1]✉

mTORC1, a central controller of cell proliferation in response to growth factors and nutrients, is dysregulated in cancer. Whereas arginine activates mTORC1, it is overridden by high expression of cytosolic arginine sensor for mTORC1 subunit 1 (CASTOR1). Because cancer cells often encounter low levels of nutrients, an alternative mechanism might exist to regulate CASTOR1 expression. Here we show K29-linked polyubiquitination and degradation of CASTOR1 by E3 ubiquitin ligase RNF167. Furthermore, AKT phosphorylates CASTOR1 at S14, significantly increasing its binding to RNF167, and hence its ubiquitination and degradation, while simultaneously decreasing its affinity to MIOS, leading to mTORC1 activation. Therefore, AKT activates mTORC1 through both TSC2- and CASTOR1-dependent pathways. Several cell types with high CASTOR1 expression are insensitive to arginine regulation. Significantly, AKT and RNF167-mediated CASTOR1 degradation activates mTORC1 independent of arginine and promotes breast cancer progression. These results illustrate a mTORC1 regulating mechanism and identify RNF167 as a therapeutic target for mTORC1-dysregulated diseases.

[1] UPMC Hillman Cancer Center, Department of Microbiology and Molecular Genetics, University of Pittsburgh, Pittsburgh, PA, USA. ✉email: gaos8@upmc.edu

The serine/threonine kinase AKT is mutated in about 10% of human cancer, which largely accounts for its onco-genicity in cancer[1]. Besides mutations in the AKT gene, the dysregulation of upstream pathways of growth factors often activates AKT in cancer cells. For examples, the dysregulation of estrogen receptor (ER), progesterone, and human epidermal growth factor 2 (HER2) leads to constitutive AKT phosphoryla-tion and activation in >80% breast cancer[2]. AKT has >100 sub-strates. Among them, three major downstream nodes, including GSK3β, FOXOs, and TSC2, mediate AKT's diverse functions in response to different stimulations[3]. The most prominent con-sequence of AKT-mediated phosphorylation of a given protein is cellular translocation (e.g., FOXOs), degradation (e.g., GSK3β and TSC2), or alteration of protein–protein interaction (e.g., TSC2). So far, AKT-mediated phosphorylation and inhibition of TSC2 has been described as the primary mechanism of AKT activation of the mammalian target of rapamycin complex 1 (mTORC1)[4,5]. A peptide screening has identified the specific sequence of AKT substrates with a minimal consensus recognition motif of R-X-R-X-X-S/T-Φ, where X is any amino acid (AA) and Φ denotes a preference for large hydrophobic residues[6].

Cytosolic arginine sensor for mTORC1 subunit 1 (CASTOR1) is a newly discovered arginine sensor and regulates mTORC1 activity in response to arginine status[7,8]. Upon arginine depri-vation, CASTOR1 interacts with and sequesters the critical positive regulator of mTORC1, the GATOR2 complex; in con-trast, arginine stimulation releases GATOR2 from CASTOR1 and subsequently activates mTORC1[7]. Interestingly, a high level of CASTOR1 protein inhibits mTORC1 activation by AAs, includ-ing arginine[7]. Of note, tumor cells often have limited access to exogenous nutrients, including AAs, glucose, and oxygen[9]. In particular, argininosuccinate synthase 1 (ASS1), the rate-limiting enzyme for endogenous arginine de novo synthesis, is silenced in up to 90% of cancer, rendering cancer cells arginine auxotrophic[10,11]. Since cancer cells have constitutively activated mTORC1, it is expected that the expression and function of CASTOR1 are inhibited by an alternative mechanism(s) rather than arginine[12]. By investigating the mechanism of Kaposi's sarcoma-associated herpesvirus (KSHV) induction of cellular transformation, we have previously reported that KSHV encodes viral microRNAs to target CASTOR1 leading to the activation of mTORC1[13]. As no specific CASTOR1 mutation associated with cancer has been described so far, how other cancer cells evade the inhibitory effect of CASTOR1 on mTORC1 in nutrient-deficient, especially AA-deficient, tumor microenvironment in other types of cancer remains unclear.

Using the kinase prediction algorithms[14], we have predicted that CASTOR1 contains a consensus AKT1 phosphorylation motif R-V-R-V-L-S14. Proteomic analysis indeed identified CASTOR1 phosphorylation at S14[15], further suggesting that CASTOR1 is a potential AKT1 substrate. Examination with the point mutation prediction algorithms revealed an increased sta-bility of CASTOR1 if S14 is mutated to a non-phosphorylatable mimic alanine (A) and a decreased stability if it is mutated to a constitutively phosphorylated mimic aspartic acid (D)[16]. These analyses imply that AKT1 might phosphorylate CASTOR1 and regulate its stability.

The phosphorylation-dependent regulation of protein stability is closely associated with protein polyubiquitination[17], a mark for their degradation via 26S proteasome. The formation of poly-ubiquitin chain conjugated to a target protein occurs in a cascade of three steps: activation, conjugation, and ligation, exerted by E1 ubiquitin-activating enzyme, E2-conjugating enzyme, and E3 ubiquitin ligase, respectively[18]. The first linkage is initiated by the binding of the C-terminal glycine in ubiquitin to the lysine in the substrate, forming an isopeptide bond. Further polyubiquitin chain can be formed by linking the glycine residue of another ubiquitin molecule to the lysine of ubiquitin bound to a sub-strate[18]. Seven lysine residues in ubiquitin are responsible for polyubiquitin formation, including K6, K11, K27, K29, K33, K48, and K63. Among them, K29-, K48-, or K63-mediated poly-ubiquitination typically triggers proteasomal degradation[19].

RING finger protein (RNF167) is a RING-type E3 ligase involved in regulating protein trafficking, localization, and degradation by directly ubiquitinating targeted substrates[20–22].

In this study, we report that a low expression level of CAS-TOR1 is correlated with poor patient survival in numerous types of cancer including breast cancer and that CASTOR1 is a sub-strate of RNF167. Furthermore, AKT-mediated phosphorylation of CASTOR1 facilitates its interaction with RNF167, leading to CASTOR1 ubiquitination and proteasome-dependent degrada-tion. Additionally, CASTOR1 phosphorylation at S14 by AKT decreases its binding affinity to GATOR2 complex. The phos-phorylation and degradation of CASTOR1 collectively release the GATOR2 complex, activate mTORC1, and promote breast cancer progression. These findings reveal a mechanism by which cancer cells overcome the suppressive effect of CASTOR1 in the nutrient-deficient tumor microenvironment and hence identify a potential therapeutic target for treating mTORC1-associated diseases, including cancer.

## Results

**RNF167 mediates K29-linked polyubiquitination and degra-dation of CASTOR1 in response to growth factors**. To reveal the environmental cue that activates mTORC1 by modulating the expression of CASTOR1, we deprived cells of either fetal bovine serum (FBS) or arginine. The kinetic analysis demonstrated that CASTOR1 protein level but not mRNA level was increased fol-lowing 16 h of FBS deprivation in 293T cells, which was corre-lated with a decreased mTORC1 activity as shown by the reduced phosphorylation level of its downstream targets S6K and 4EBP1 (Fig. 1a and Supplementary Fig. 1a). As expected, the level of AKT activation was significantly reduced, which was noticeable at as early as 2 h but more obvious after 16 h following FBS depri-vation (Fig. 1a). Hence, the level of CASTOR1 protein inversely trailed that of AKT activation following FBS deprivation. In contrast, arginine deprivation for as short as 15 min in 293T cells resulted in decreased mTORC1 activity (Fig. 1b). However, there were only marginal fluctuations of activated AKT and CASTOR1 protein levels before the first 4 h of arginine deprivation. The decreased mTORC1 activity was likely due to the released argi-nine inhibitory effect on CASTOR1[7,8]. Extended arginine depri-vation for >8 h enhanced AKT activation as a result of the feedback effect of mTORC1 inhibition[23,24], which was correlated with a slight decrease of CASTOR1 protein level as well as a slight decrease of mRNA level (Fig. 1b and Supplementary Fig. 1b). There was no increase of mTORC1 activity in these later time points despite the increased AKT activity and reduced CASTOR1 protein level. This was likely due to its already low mTORC1 activity as well as the requirement of arginine for its activation. In agreement with the results in 293T cells, deprivation of either FBS or arginine inactivated mTORC1 in ER+ breast cancer cell lines MCF7 and T47D albeit their response kinetics varied (Supple-mentary Fig. 1c–f). FBS deprivation inactivated AKT at as early as 15 min and CASTOR1 protein level started to increase by 8 h following FBS deprivation (Supplementary Fig. 1c, d). Thus, similar to 293T cells, the level of CASTOR1 protein inversely trailed that of AKT activation following FBS deprivation in these cells. Following arginine deprivation, marginal fluctuations of activated AKT and CASTOR1 protein levels were also observed within the first 1 h (Supplementary Fig. 1e, f). However, a

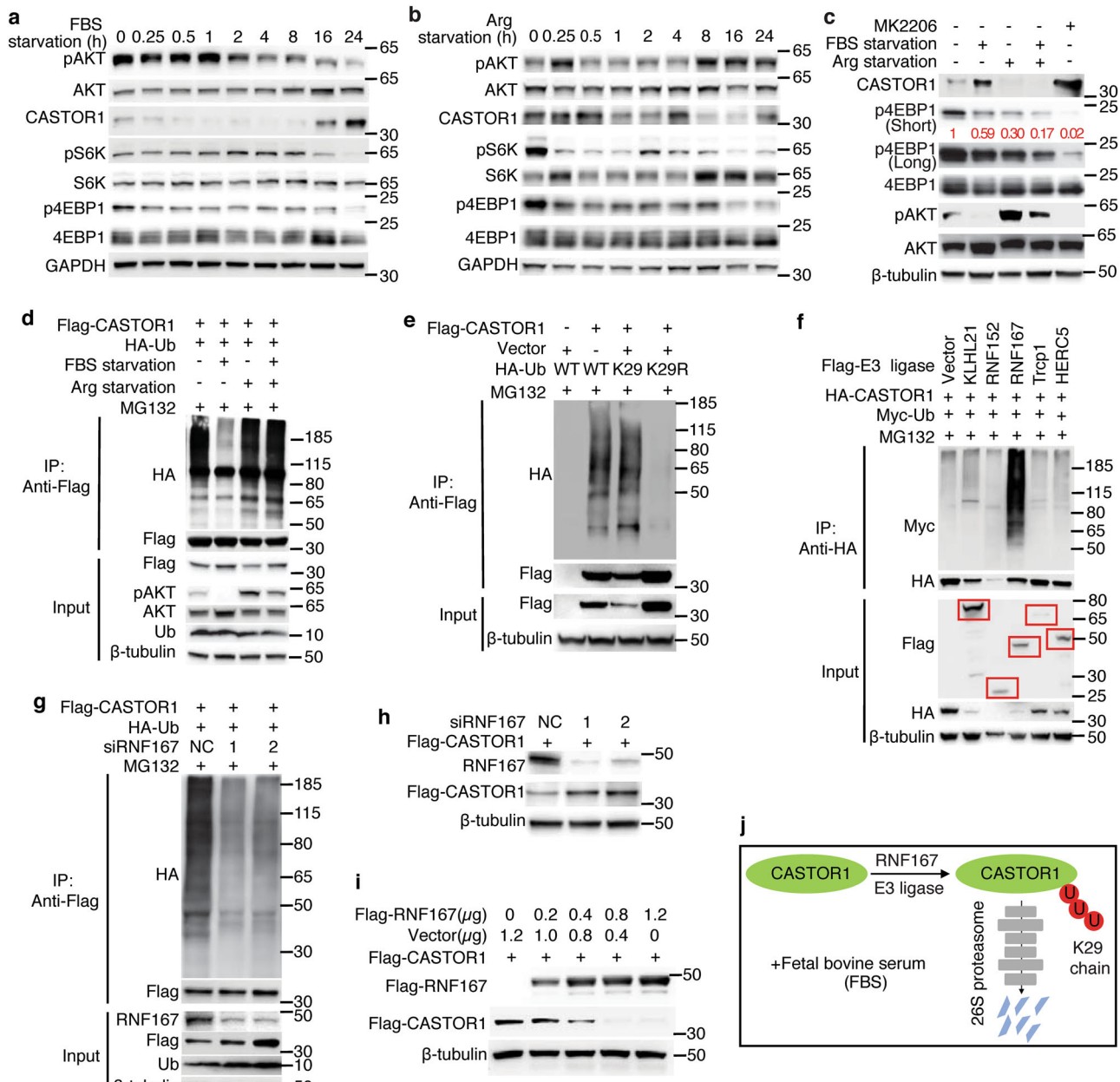

**Fig. 1 RNF167 mediates K29-linked polyubiquitination and degradation of CASTOR1 in response to growth factors. a** Kinetics of CASTOR1 protein level and activation status of AKT and mTORC1 following fetal bovine serum (FBS) deprivation in 293T cells. **b** Kinetics of CASTOR1 protein level and activation status of AKT and mTORC1 following arginine deprivation in 293T cells. **c** CASTOR1 protein level and activation status of AKT and mTORC1 following deprivation of FBS, arginine, or both or treatment with AKT inhibitor MK2206 in 293T cells. FBS or arginine deprivation or AKT inhibitor treatment was carried out for 24 h. **d** CASTOR1 ubiquitination status following deprivation of FBS, arginine, or both. **e** CASTOR1 was labeled by K29-linked polyubiquitination. An ubiquitin mutant K29 contained only the K29 lysine residue was sufficient to cause CASTOR1 polyubiquitination while mutation of K29 (K29R) abolished CASTOR1 polyubiquitination. **f**, **g** Ectopic expression of RNF167 increased (**f**), whereas knockdown of RNF167 decreased (**g**) CASTOR1 ubiquitination. **h**, **i** RNF167 knockdown increased (**h**), whereas RNF167 overexpression decreased (**i**) CASTOR1 protein level. **j** Schematic depiction of the K29-marked polyubiquitination and degradation of CASTOR1 protein by RNF167 in response to FBS. Blots in **a–i** are representatives of $n = 3$ independent experiments. Source data are provided in Source data file.

decrease of activated AKT was observed between 2 and 4 h, which led to a slight increase of CASTOR1 protein level at 8 and 16 h. The mTORC1 activity was not further decreased at these time points, which was likely due to its already low level. Similar to 293T cells, we observed enhanced AKT activation after 16 h as a result of the feedback effect of mTORC1 inhibition, which led to a slight decrease of CASTOR1 protein level at 24 h following arginine deprivation (Supplementary Fig. 1e, f). Only a slight increase of mTORC1 activity was observed at this time point,

which again indicated the essential role of arginine in mTORC1 activation. Intriguingly, FBS deprivation slightly increased while arginine deprivation dramatically increased CASTOR1 mRNA level in these cells (Supplementary Fig. 1g–j).

The above results showed a negative correlation of AKT activation with the CASTOR1 protein level, which was strongly regulated by FBS deprivation but only marginally regulated by arginine deprivation, suggesting an important regulatory role of growth factors in the CASTOR1 protein level. Treatment with

AKT inhibitor MK2206 in 293T cells upregulated CASTOR1 protein but not mRNA level and decreased mTORC1 activation, mimicking FBS deprivation (Fig. 1c and Supplementary Fig. 1k). Because mTORC1 could be responsive to other nutrients such as leucine present in the medium, we further examined the effect of leucine deprivation on CASTOR1 protein. Similar to arginine deprivation, chronic leucine deprivation activated AKT and reduced CASTOR1 mRNA and protein levels and mTORC1 activity (Supplementary Fig. 1k, l). Interestingly, an S6K1 inhibitor that decreased the pS6K but not p4EBP1 level failed to activate AKT and reduce CASTOR1 protein level (Supplementary Fig. 1l). Together these results suggest the involvement of a regulatory role of CASTOR1 in the AKT-mTORC1 loop.

**AKT1 phosphorylation of CASTOR1 promotes RNF167-mediated ubiquitination and degradation of CASTOR1.** Since our results suggested that the CASTOR1 protein level was strongly regulated by FBS, potentially through AKT activation, we further examined the mechanism mediating CASTOR1 degradation. Consistent with the observed CASTOR1 protein level, FBS deprivation reduced CASTOR1 ubiquitination, while arginine deprivation had no noticeable effect (Fig. 1d). FBS re-stimulation after deprivation reversed the effect, restoring CASTOR1 ubiquitination, which was correlated with the reduced CASTOR1 protein level (Supplementary Fig. 2a). Together, these results confirmed that arginine did not significantly affect CASTOR1 ubiquitination and protein level but FBS targeted CASTOR1 for ubiquitination and proteasome-dependent degradation.

Covalent conjugation of ubiquitin is a key step in proteasome-mediated degradation of target proteins[19]. CASTOR1 was only labeled by wild-type (WT) ubiquitin or K29 ubiquitin, a ubiquitin mutant containing only the K29 lysine, but not by K48 and K63 ubiquitin (Fig. 1e and Supplementary Fig 2b, c). Mutation of K29 ubiquitin (K29R) abolished CASTOR1 ubiquitination (Fig. 1e). These results indicated that K29 ubiquitin was essential and sufficient to mediate CASTOR1 ubiquitination.

To identify the E3 ubiquitin ligase(s) that might regulate CASTOR1 polyubiquitination and degradation, we screened a panel of E3 ubiquitin ligases implicated in mTORC1 regulation[25]. Although ectopic expression of numerous E3 ubiquitin ligases decreased CASTOR1 protein level (Supplementary Fig. 2d), only RNF167 increased CASTOR1 ubiquitination (Fig. 1f and Supplementary Fig. 2e). Consistently, knockdown of RNF167 decreased CASTOR1 ubiquitination and increased CASTOR1 protein level (Fig. 1g, h) while overexpression of RNF167 decreased CASTOR1 protein level in a dose-dependent manner (Fig. 1i). Neither overexpression nor knockdown of RNF167 had notable effect on the CASTOR1 mRNA level (Supplementary Fig. 2f, g). Additionally, treatment with MG132 partially rescued RNF167-mediated downregulation of CASTOR1 protein (Supplementary Fig. 2h). These results support a model that RNF167 targets CASTOR1 for ubiquitination and proteasome-dependent degradation (Fig. 1j).

By providing growth factors, FBS activates numerous kinases, which could be the reason that it regulates CASTOR1 level. Since the effect of AKT inhibitor MK2206 on CASTOR1 protein level was the same as FBS starvation (Fig. 1c), we used kinase prediction algorithms and identified a consensus AKT1 phosphorylation site on CASTOR1 with a motif of R-V-R-V-L-S14. Proteomic analysis indeed identified CASTOR1 phosphorylation at S14[14,15], suggesting that AKT1 might directly phosphorylate CASTOR1. Indeed, CASTOR1 interacted with both ectopically expressed AKT1 and endogenous AKT and preferentially bound to AKT1 kinase domain (Supplementary Fig. 3a–f). An antibody specific to the AKT phosphorylation consensus motif (R-X-R-X-

X-pS/T) detected a strong signal in the WT HA- or Flag-CASTOR1 protein expressed in 293T cells, confirming that CASTOR1 was phosphorylated at the physiological condition (Fig. 2a, b). Importantly, the level of CASTOR1 phosphorylation at S14 was positively correlated with AKT activation, which was increased following deprivation of arginine or leucine but decreased following deprivation of FBS or all AAs (Fig. 2a). Furthermore, CASTOR1 protein level was negatively correlated with CASTOR1 phosphorylation at S14 (Fig. 2a), suggesting that AKT mediated CASTOR1 phosphorylation at S14 to target its degradation. In agreement with these results, an alanine substitution at S14 (Flag-CASTOR1 S14A), which generated a phosphorylation dead mutant, and AKT inhibitor MK2206 significantly reduced the specific phosphorylation of the AKT motif (Fig. 2b, c), hence confirming AKT-mediated phosphorylation of CASTOR1 at S14. Alignment of CASTOR1 protein sequences from human with other vertebrates revealed that the CASTOR1 R-X-R-X-X-S14 motif was highly conserved (Supplementary Fig. 3g). As expected, AKT interacted with and phosphorylated CASTOR1 at the AKT phosphorylation motif in rat metanephric mesenchymal precursor (MM) cells and KSHV-transformed MM (KMM) cells (Supplementary Fig. 3h)[26].

We performed in vitro kinase assay to confirm AKT direct phosphorylation of CASTOR1. Purified glutathione S-transferase (GST)-AKT1 recombinant protein efficiently phosphorylated purified GST-tagged CASTOR1 (GST-CASTOR1) recombinant protein only in the presence of ATP, which was abolished by AKT inhibitor MK2206 (Fig. 2d and Supplementary Fig. 3i). Interestingly, Flag-CASTOR1 S14D, a mimic of constitutively phosphorylated mutant, had a much higher affinity to AKT1 than Flag-CASTOR1 WT and Flag-CASTOR1 S14A (Supplementary Fig. 3j–m), suggesting possible CASTOR1 conformational changes following phosphorylation. A similar observation that the AKT3-Ago2 interaction was enhanced following AKT3 phosphorylation of Ago2 at S387 was previously reported[27]. Collectively, these results demonstrated that AKT directly bound to and phosphorylated CASTOR1.

As phosphorylation is intimately linked to protein ubiquitination and degradation[17], we examined the consequence of AKT1-mediated CASTOR1 phosphorylation and observed that myristoylated constitutively active AKT1 (myr-HA-AKT1) decreased the CASTOR1 protein level in a dose-dependent manner (Fig. 2e). Neither the kinase-dead AKT1 mutant (K179M) nor the AKT1 PH domain had any effects while overexpression of the AKT1 kinase domain alone was sufficient to reduce the CASTOR1 protein level albeit to a lesser degree than the WT AKT1 (Supplementary Fig. 4a, b)[28]. Hence, AKT-mediated CASTOR1 downregulation required its kinase activity. Neither the WT AKT1, AKT1 PH, and kinase domains nor the kinase-dead mutant affected the CASTOR1 mRNA level (Supplementary Fig. 4c–e). Consistently, AKT1 silencing was sufficient to inhibit pan AKT activity and increased the CASTOR1 protein level (Fig. 2f) but had no effect on the CASTOR1 mRNA expression (Supplementary Fig. 4f).

To test whether AKT1 regulated CASTOR1 stability, we first co-transfected cells with both Flag-CASTOR1 WT and myr-HA-AKT1, then treated them with de novo protein synthesis inhibitor cycloheximide (CHX), and observed faster degradation of CASTOR1 protein in cells expressing myr-HA-AKT1 than the vector control (Supplementary Fig. 4g). Treatment with proteasome inhibitor MG132 increased the accumulation of CASTOR1 protein in cells expressing myr-HA-AKT1 but only had a marginal effect on cells expressing the vector control (Supplementary Fig. 4h). Furthermore, overexpression of myr-HA-AKT1 but not AKT1 mutant (K179M) enhanced, whereas knockdown of AKT1 reduced CASTOR1 ubiquitination (Fig. 2g, h). Together,

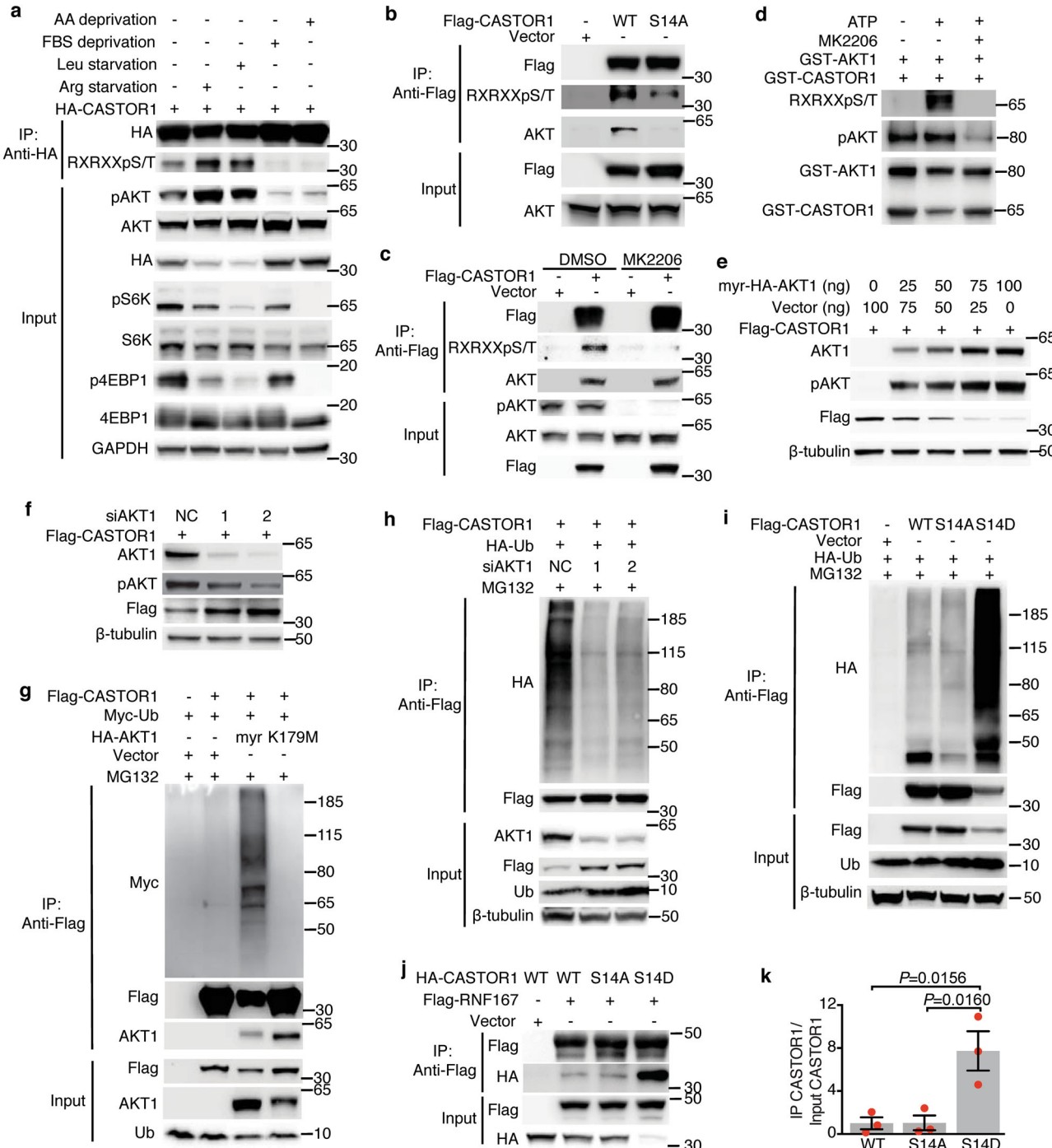

**Fig. 2 AKT1 phosphorylation of CASTOR1 promotes RNF167-mediated ubiquitination and degradation of CASTOR1. a** Deprivation of arginine or leucine activated AKT and increased CASTOR1 phosphorylation at S14, whereas deprivation of FBS or total amino acids inactivated AKT and reduced CASTOR1 phosphorylation at S14. **b** CASTOR1 S14 phosphorylation was markedly reduced following alanine substitution (S14A). **c** AKT inhibition abolished CASTOR1 phosphorylation at S14 in vivo. **d** Recombinant AKT1 protein directly phosphorylated CASTOR1 protein in vitro. **e**, **f** AKT overexpression increased (**e**) while AKT1 knockdown decreased (**f**) CASTOR1 degradation. **g**, **h** AKT1 overexpression increased (**g**) and AKT1 knockdown decreased (**h**) CASTOR1 ubiquitination. **i** CASTOR1 S14D had increased ubiquitination level compared to WT and S14A. **j**, **k** Phosphorylation of CASTOR1 at S14 significantly increased its affinity to RNF167 (**j**), and quantifications of results from three independent experiments are presented (**k**). For **k**, data are presented as mean values ± SEM and P value was calculated by one-way ANOVA followed by Tukey post hoc test (*n* = 3 independent experiments). Blots in **a–c**, **e–j** are representatives of *n* = 3 independent experiments, and blots in **d** are representatives of *n* = 2 independent experiments. Source data are provided in Source data file.

these results confirmed that AKT1 targeted CASTOR1 for ubiquitination and proteasome-dependent degradation.

We constructed 293T cells stably expressing Flag-CASTOR1 WT, S14A, or S14D and observed that cells expressing Flag-

CASTOR1 S14D had lower protein level than those expressing CASTOR1 WT and S14A despite there being no significant change at the mRNA level (Supplementary Fig. 5a). Indeed, treatment with CHX reduced while treatment with MG132

increased S14D protein level but had minimal effects on WT and S14A (Supplementary Fig. 5b–d). Accordingly, the level of ubiquitination was significantly increased for S14D protein compared to those of WT and S14A proteins (Fig. 2i and Supplementary Fig. 5e). These results demonstrated that AKT1 phosphorylation of CASTOR1 at S14 resulted in its ubiquitination and degradation.

To clarify the link between AKT1-mediated phosphorylation and RNF167-mediated ubiquitination of CASTOR1, we examined the effect of CASTOR1 phosphorylation on CASTOR1-RNF167 interaction. CASTOR1 S14D had a much stronger affinity to RNF167 and a higher level of ubiquitination than WT or S14A had (Fig. 2i–k and Supplementary Fig. 5e), indicating that AKT-mediated phosphorylation promoted CASTOR1 degradation by specifically enhancing the CASTOR1–RNF167 interaction. Collectively, these results support a model that AKT1 phosphorylation of CASTOR1 at S14 enhances RNF167-targeting ubiquitination and degradation of CASTOR1 protein.

Examination of CASTOR1 with the Ubisite and UbPreb program identified numerous lysine residues as putative ubiquitination sites, including K61, K96, and K213 (Supplementary Fig. 6a)[29]. Whereas mutation of one or two of these sites to arginine in CASTOR1 S14D failed to stabilize the protein, mutation of all three sites to arginine (3KR) significantly blocked CASTOR1 ubiquitination and degradation (Supplementary Fig. 6b–d). Importantly, while all single and double lysine mutants of CASTOR1 S14D remained sensitive to RNF167-mediated downregulation, the 3KR mutant was resistant (Supplementary Fig. 6e), indicating that RNF167 catalyzed CASTOR1 ubiquitination at multiple lysines.

**High CASTOR1 protein level overrides arginine activation of mTORC1**. Next, we assessed the downstream effects of AKT1-mediated phosphorylation and RNF167-targeting degradation of CASTOR1 protein. Consistent with the previous report[7], a high level of CASTOR1 protein rendered cells insensitive to arginine-mediated activation of mTORC1 in 293T cells (Fig. 3a). To determine whether CASTOR1 regulates mTORC1 activation in physiological conditions, we examined CASTOR1 protein levels in different types of cells and tested their sensitivities to arginine (Fig. 3b). HeLa cells, which had almost no detectable CASTOR1 protein expression, were resistant to mTORC1 inactivation by arginine deprivation (80 min) as well as mTORC1 activation by 10-min arginine re-stimulation following 50-min arginine deprivation (i.e., arginine-mediated mTORC1 activation, Fig. 3b–d). These results indicated that mTORC1 was constitutively activated when CASTOR1 protein expression was completely silenced and that these cells were no longer responsive to arginine. MCF7 cells, which had a low CASTOR1 protein level, were responsive to arginine-mediated mTORC1 activation (Fig. 3b, e). In contrast, cells with high endogenous CASTOR1 protein levels including human lobar bronchial epithelial cells (HLBEC), human small airway epithelial cells (HSAEC), and T47D were not responsive to arginine-mediated mTORC1 activation (Fig. 3b, e, f), suggesting CASTOR1's strong suppressive role in mTORC1 activity in these cells. Under this condition, no CASTOR1 protein level change was observed in these cells. In agreement with these results, silencing of CASTOR1 in T47D cells, which had a high endogenous CASTOR1 protein level, was sufficient to strongly activate mTORC1, further supporting CASTOR1's direct regulatory role in mTORC1 activity (Fig. 3b, g). Together, these results supported the notion that a high CASTOR1 protein level overrode arginine activation of mTORC1 at physiological conditions and the mTORC1 activity was tightly regulated by CASTOR1 instead of the arginine status when CASTOR1 was expressed at a high level (Fig. 3h).

**RNF167-mediated ubiquitination and AKT-mediated phosphorylation of CASTOR1 release mTORC1 inactivation**. Mechanistically, binding of CASTOR1 to MIOS, the core component of GATOR2 complex, was positively correlated with the CASTOR1 protein level, further demonstrating that mTORC1 activation was regulated by the CASTOR1 protein level in addition to arginine (Fig. 4a). As expected, ectopic expression of RNF167 degraded CASTOR1 and activated mTORC1 regardless of the presence or absence of arginine (Fig. 4b). In fact, cells with overexpression of RNF167 became insensitive to arginine-mediated mTORC1 activation (Fig. 4b), affirming the essential role of RNF167 and regulation of CASTOR1 protein level in the control of mTORC1 activation. As expected, myr-HA-AKT1 but not its kinase-dead mutant K179M decreased CASTOR1 protein level and hence its binding to MIOS, resulting in increased mTORC1 activation (Fig. 4c). The consensus mechanism of AKT-mediated activation of mTORC1 is by suppressing TSC2[5]. However, our results suggested that AKT might also activate mTORC1 by targeting CASTOR1 for degradation. To dissect AKT's independent effects on CASTOR1 and TSC2 in regulating mTORC1, we performed knockdown of TSC2 and examined CASTOR1 expression and mTORC1 activity in the presence or absence of FBS. As expected, FBS deprivation reduced AKT activation and increased CASTOR1 protein level leading to mTORC1 inactivation in controlled cells (Fig. 4d). Silencing of TSC2 had no effect on AKT activation and CASTOR1 protein level but was sufficient to activate mTORC1. However, mTORC1 was still inactivated by the increased CASTOR1 protein level following FBS deprivation in the TSC2-silencing cells (Fig. 4d). These results indicated that AKT activated mTORC1 through two independent pathways, by reducing CASTOR1 protein level and by suppressing TSC2.

Since CASTOR1 S14D was constitutively phosphorylated and hence was prone to degradation, whereas CASTOR1 S14A was non-phosphorylatable and resistant to degradation, we utilized these constructs to assess the effect on mTORC1 activation. Consistently, the protein level was lower, which led to a lower pull down yield in co-immunoprecipitation, for S14D than for WT and S14A (Fig. 4e, f). Furthermore, S14D binding to MIOS was significantly weaker than that of WT or S14A even after taking into consideration its lower protein level and lower pull down efficiency in co-immunoprecipitation (Fig. 4e–g). Hence, a lower protein level and a lower affinity to MIOS might lead to a more robust mTORC1 activation for S14D than for WT and S14A (Fig. 4e, g). These differences persisted even with arginine concentration reaching 50 μM indicating that the combined effects of AKT1 phosphorylation and RNF167-targeting degradation had a stronger role than arginine inhibition of CASTOR1 in regulating mTORC1 activation, particularly at a condition with a low concentration of arginine, which is common in tumor microenvironment (Fig. 4h).

**RNF167-mediated ubiquitination and AKT1-mediated phosphorylation of CASTOR1 promote breast cancer progression**. We examined the prognostic value of CASTOR1 mRNA expression in cancer using the TCGA database. Consistent with CASTOR1's inhibitory function on mTORC1 and tumor-suppressive role[13,30], a lower CASTOR1 expression level was correlated with overall poor survival in pan-cancer analyses (Supplementary Fig. 7a, b). At least 10 types of cancer showed a strong negative correlation, including breast invasive carcinoma, brain lower grade glioma (LGG), skin cutaneous melanoma

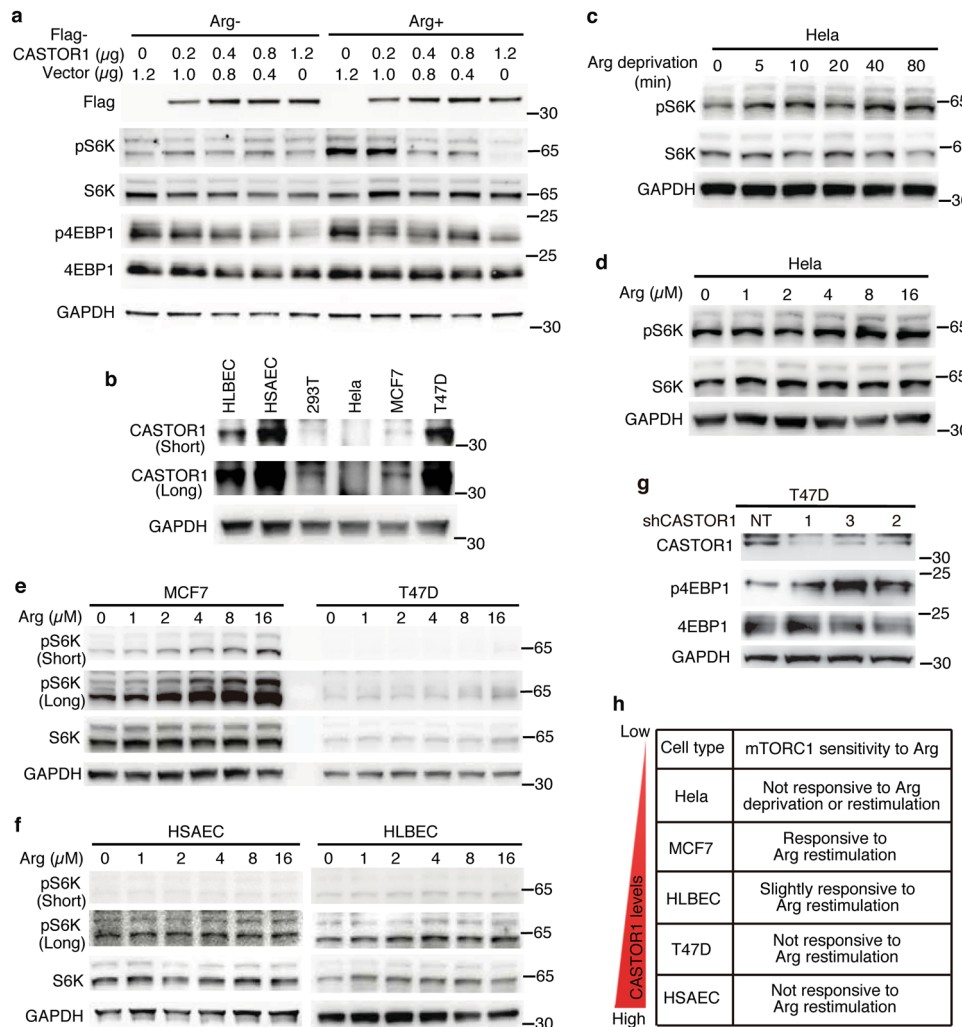

**Fig. 3 High CASTOR1 protein level overrides arginine activation of mTORC1 in physiological conditions. a** Response of mTORC1 activation to CASTOR1 overexpression in a dose-dependent manner with and without the presence of arginine in 293T cells. High level of CASTOR1 overrode arginine-mediated mTORC1 activation. **b** CASTOR1 protein expression levels in multiple cell types, including human lobar bronchial epithelial cells (HLBEC), human small airway epithelial cells (HSAEC), 293T, HeLa, and breast cancer cell lines MCF7 and T47D. **c, d** HeLa cells, which had almost no detectable CASTOR1 protein and a high level of constitutively activated mTORC1, was minimally responsive to arginine regulation of mTORC1, including arginine deprivation for 80 min (**c**) and re-stimulation for 10 min following arginine deprivation for 50 min (**d**). **e** MCF7 cells were more responsive than T47D cells to arginine-mediated mTORC1 activation, which was inversely correlated with their CASTOR1 protein levels (**b**). **f** Cells with high endogenous CASTOR1 protein levels including HSAEC and HLBEC (**b**) were not responsive to arginine-mediated mTORC1 activation. **g** CASTOR1 knockdown in T47D cells, which had a high level of endogenous CASTOR1 protein (**b**), activated mTORC1. **h** Summary of the relative endogenous CASTOR1 protein expression levels in different types of cells and their responsiveness to arginine regulation of mTORC1. Blots in **a–g** are representative of $n = 3$ independent experiments. Source data are provided in Source data file.

(SKCM), head and neck squamous cell carcinoma, cervical squamous cell carcinoma and endocervical adenocarcinoma, lung adenocarcinoma, liver hepatocellular carcinoma (LIHC), pancreatic adenocarcinoma, glioblastoma multiforme (GBM), and acute myeloid leukemia (LAML) (Supplementary Fig. 7c); of these, high RNF167 expression predicted a poor prognosis in GBM, LAML, SKCM, LGG, and LIHC (Supplementary Fig. 7d).

Breast cancer represents 12% of cancer diagnosed and is a major life threat for women in the United States[2]. We found a high RNF167 expression level in breast tumors compared to the adjacent normal tissues (Supplementary Fig. 7e). Furthermore, a lower CASTOR1 expression level (Supplementary Fig. 7f, g) and a higher RNF167 expression level (Supplementary Fig. 7h, i) were correlated with poor survival in ER+ and HER2+ breast cancer, respectively. In two pairs of ER+ and HER2+ breast cancer cell lines, we found an inverse correlation of activated AKT level with CASTOR1 protein level (Supplementary Fig. 8a). AKT interacted with CASTOR1 in MCF7 cells (Supplementary Fig. 8b). Silencing of AKT1 and AKT inhibitor MK2206 enhanced exogenous and endogenous CASTOR1 protein levels in these cells, respectively (Supplementary Fig. 8c, d). Consistently, overexpression of myr-HA-AKT1 but not the AKT kinase dead mutant K179M in MCF7 and T47D cells resulted in a dose-dependent reduction in CASTOR1 protein level (Supplementary Fig. 8e, f).

Consistent with 293T cells, we found that the affinity to exogenous and endogenous RNF167 was stronger for CASTOR1 S14D than for WT and S14A in ER+ MCF7 and T47D cells, respectively (Supplementary Fig. 9a–c). Likewise, RNF167 overexpression decreased, whereas RNF167 knockdown increased the CASTOR1 protein level in MCF7 cells (Supplementary Fig. 9d, e).

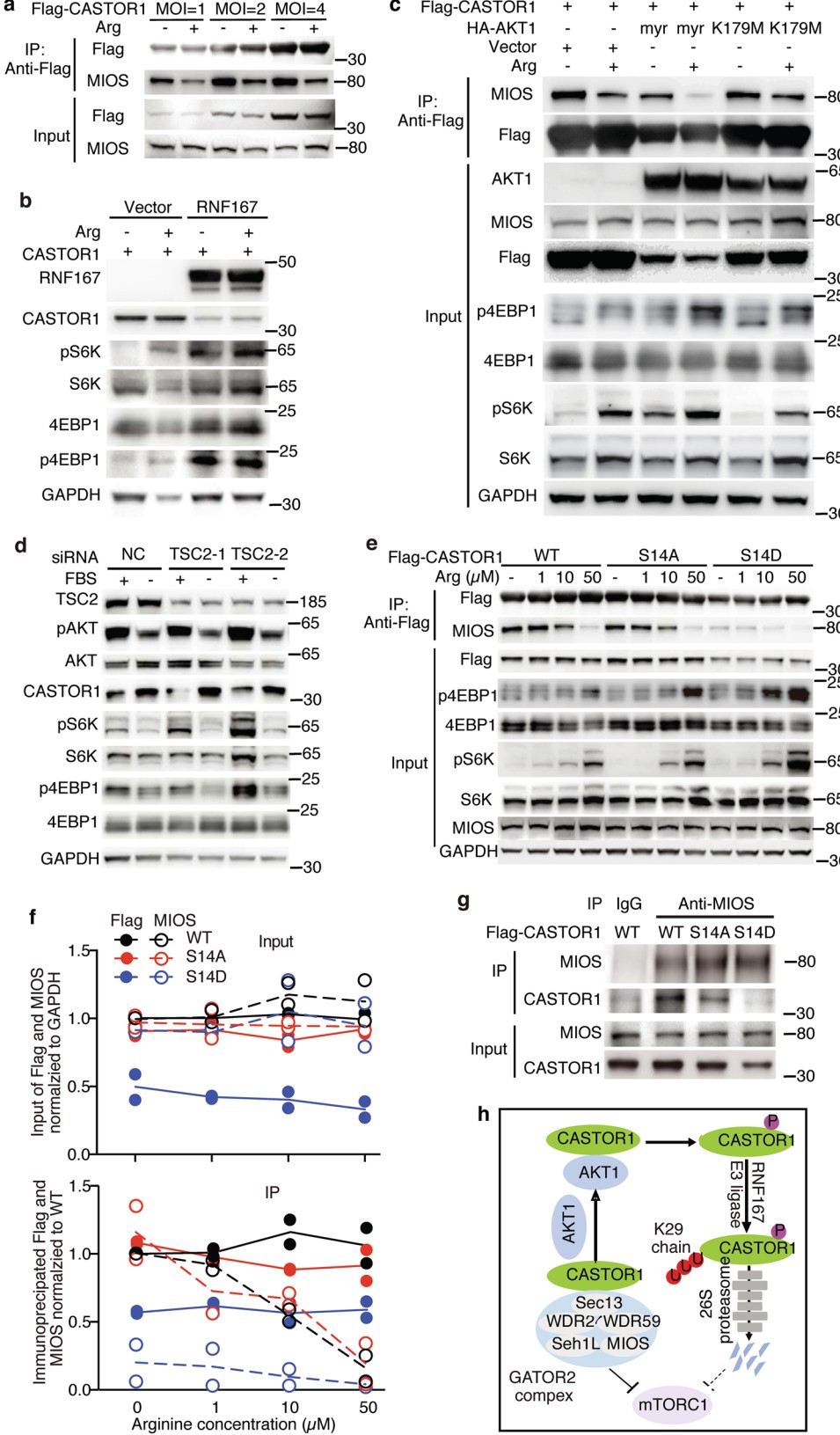

Together these results indicated that, similar to 293T cells, the CASTOR1 protein level was also regulated by AKT and RNF167 in breast cancer cells.

To examine the importance of AKT1-mediated phosphorylation and RNF167-mediated degradation of CASTOR1 in breast cancer cells, we overexpressed Flag-CASTOR1 WT, S14A, and

S14D in HCC1569, MCF7, and T47D cells. CASTOR1 S14D had much lower expression level than WT and S14A had in all the three cell lines examined (Supplementary Fig. 10a–c), indicating that S14D also had faster turnover in breast cancer cells. Importantly, ectopic expression of both CASTOR1 WT and S14A significantly inhibited mTORC1, whereas CASTOR1 S14D

**Fig. 4 AKT-mediated phosphorylation and RNF167-mediated ubiquitination of CASTOR1 release mTORC1 inactivation by CASTOR1. a** CASTOR1 bound to MIOS in a dose-dependent manner. **b** Overexpression of RNF167 decreased CASTOR1 protein level and activated mTORC1 with and without the presence of arginine. **c** Overexpression of a myristoylated constitutively active AKT1 (myr) but not the kinase-dead AKT1 mutant (K179M) reduced CASTOR1 protein level, decreased its binding to MIOS, and activated mTORC1. **d** AKT regulated mTORC1 activity by suppressing CASTOR1 was independent of TSC2. **e–g** CASTOR1 S14D had a weaker binding to MIOS shown by CASTOR1 co-immunoprecipitation (co-IP) of MIOS (**e**) and reversed MIOS co-IP of CASTOR1 (**g**), hence S14D had a less inhibitory effect on mTORC1 than WT and S14A had (**e**, **g**), and quantification of results from two independent experiments shown in **f**. **h** An illustration depicting that AKT phosphorylation and RNF167 ubiquitination of CASTOR1 reverse CASTOR1 inactivation of mTORC1. Blots in **a–d** and **g** are representatives of $n = 3$ independent experiments, and blots in **e** are representatives of $n = 2$ independent experiments. Source data are provided in Source data file.

showed a much less inhibitory effect, confirming the fine-tuning of mTORC1 signaling pathway through CASTOR1 phosphorylation and degradation in breast cancer cells (Supplementary Fig. 10a–c). Consistent with the mTORC1 activity, the proliferation and colony formation in softagar of breast cancer cells were significantly decreased by CASTOR1 WT and S14A, whereas CASTOR1 S14D had a less effect (Fig. 5a, b and Supplementary Fig. 10d–f). In T47D cells, which had a high endogenous level of CASTOR1 protein, silencing of CASTOR1 activated mTORC1 and significantly increased the colony-formation efficiencies in softagar (Figs. 3g and 5c). Moreover, overexpression of CASTOR1 WT and S14A had a stronger effect than S14D had in inhibiting cell cycle progression in MCF7 and HCC1569 cells (Supplementary Fig. 10g–j). None of the CASTOR1 constructs had any significant effect on apoptosis (Supplementary Fig. 10h, k, l), which recapitulated the characteristic phenotype of mTORC1 inhibition.

We then subcutaneously engrafted MCF7 cells that were transduced with a vector control, Flag-tagged CASTOR1 WT, S14A, or S14D into both flanks of nude mice. Ectopic expression of CASTOR1 WT and S14A significantly inhibited tumor growth in vivo, whereas S14D had a relatively less effect (Fig. 5d–f and Supplementary Fig. 11a). Additionally, mice injected with cells expressing CASTOR1 WT and S14A had higher survival rates than those of expressing vector control and S14D (Fig. 5g). Consistently, silencing of CASTOR1 in T47D cells promoted tumor growth in vivo and shortened the overall survival compared to a scrambled control group (Fig. 5h–j and Supplementary Fig. 11b). Taken together, these results revealed that AKT-mediated phosphorylation and RNF167-dependent ubiquitination led to a decreased CASTOR1 protein level in breast cancer cells, resulting in enhanced mTORC1 activation, cell proliferation, and tumorigenesis.

## Discussion

Here we report a general mechanism of AKT-mediated phosphorylation at S14 and RING-type E3 ligase RNF167-mediated ubiquitination at multiple lysine residues of CASTOR1 leading to its proteasome-dependent degradation and consequently mTORC1 activation. The AKT phosphorylation site in CASTOR1 is present in other vertebrate species analyzed, indicating its conserved function. Mutation of this site into a constitutively phosphorylated mutant (S14D) increases its interaction with AKT, suggesting a possible conformation change and a feed-forward negative AKT regulatory mechanism of the CASTOR1 protein. We have shown that the CASTOR1 lysines, i.e., K61, K96, and K213, are marked by K29-linked polyubiquitination. Intriguingly, the constitutively phosphorylated S14D mutant has a significantly higher affinity to RNF167, explaining its faster ubiquitination and degradation, and a significantly lower affinity to MIOS. Hence, AKT-mediated CASTOR1 phosphorylation results in reduced CASTOR1 protein level and inhibition of the GATOR2 complex, both contributing to mTORC1 activation. This mechanism remains functional even after TSC2 knockdown

indicating the presence of a TSC2-independent but CASTOR1-dependent pathway of AKT-mediated mTORC1 activation. Importantly, by manipulating extracellular nutrients such as FBS and arginine in several types of cells, we have shown that this mechanism of AKT-mediated CASTOR1 degradation and mTORC1 activation is functional in physiological conditions.

mTORC1 activation is tightly regulated occurring in a cascade fashion initiated by AA-mediated mTORC1 translocation to lysosomes followed by AKT-induced Rheb phosphorylation of mTOR[12,31]. So far, several AA sensors including Sestrin2, SLC39A9, TM4SF5 and SAMTOR are known to modulate mTORC1 activity in response to AA status[32–35]. CASTOR1 is a newly discovered arginine sensor, which interplays with arginine to modulate mTORC1 signaling pathway[7,8]. Hence, our findings reveal a crosstalk between two previously independent signaling pathways, i.e., the growth factor-dependent AKT and arginine-regulated CASTOR1 signaling pathways, which fine-tunes mTORC1 activation. This regulatory mechanism is likely essential for controlling the homeostasis and proliferation of normal cells. In normal cells that are quiescent or at a low proliferating rate, AKT is inactivated, leading to upregulated CASTOR1, mTORC1 inactivation, and a decreased uptake of nutrients including arginine, which would have a minimal effect on CASTOR1's function and mTORC1 activation (Supplementary Fig. 12a). In hyperproliferating normal cells such as stimulated immune cells, a higher level of AKT activation would lead to a lower level of CASTOR1, an increased level of mTORC1 activation, and a higher level of uptake of nutrients including arginine, which would also inhibit CASTOR1 function, resulting in maximal mTORC1 activation (Supplementary Fig. 12b).

The mTORC1 pathway is often dysregulated in cancer, which is critical for the progression of cancer[12,13,25,31]. While CASTOR1's mTORC1 inhibitory function is negated by arginine, a high level of CASTOR1 protein evades the effect of arginine and prevents arginine-mediated mTORC1 activation (Fig. 3)[7]. Furthermore, cancer cells often survive in an environment with low nutrients including a low level of arginine[9]. Hence, it is expected that cancer cells would have evolved specific mechanisms to counter CASTOR1's inhibitory effect on mTORC1 in nutrient-deficient tumor microenvironment. In KSHV-transformed cells, KSHV-encoded miRNAs downregulate CASTOR1 to activate mTORC1[13]. In other types of cancer, the AKT pathway is persistently activated as a result of mutation of AKT itself or its upstream pathways of growth factors[36], which would phosphorylate CASTOR1 leading to its ubiquitination and degradation, and activation of mTORC1 regardless of the presence of high or low level of arginine (Supplementary Fig. 12c, d). Thus cancer cells at least partially utilize constitutively active AKT to inhibit CASTOR1's function leading to constitutive mTORC1 activation.

While no consistent association of CASTOR1 mutation with any types of cancer has been identified so far, we have found that a lower mRNA expression level of CASTOR1 predicts a poor prognosis in 10 types of cancer (Supplementary Fig. 7c). Importantly, a lower mRNA expression level of RNF167 predicts

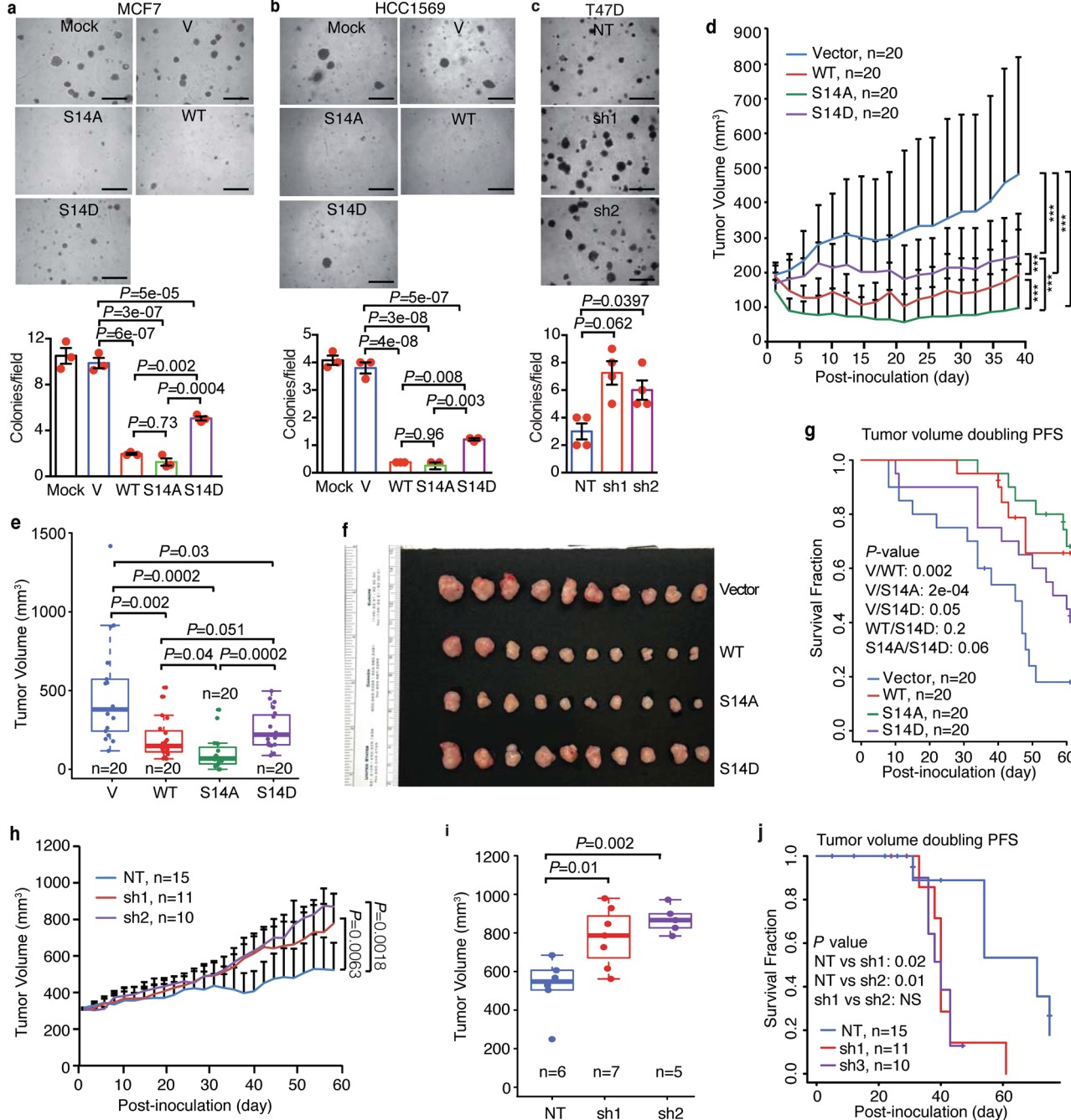

**Fig. 5 RNF167-mediated ubiquitination and AKT1-mediated phosphorylation of CASTOR1 promotes breast cancer progression. a, b** Weaker suppression of colony formation of ER+ (**a**) and HER2+ (**b**) breast cancer cells in softagar by CASTOR1 S14D than WT and S14A. **c** CASTOR1 silencing enhanced colony formation in softagar of T47D cells. **d–g** Overexpression of CASTOR1 S14D had a less suppressive effect on breast tumor growth and a lower extended animal survival rate than WT and S14A had in a breast cancer xenograft model; the tumor volumes at the indicated time point post-inoculation were measured (**d**); the tumor volumes of the last time point were compared (**e**), and the actual tumors (**f**) and the survival rates (**g**) are shown. **h–j** CASTOR1 knockdown promoted tumor growth and shortened animal survival rate. The tumor volumes at the indicated time point post-inoculation were measured (**h**); and the tumor volumes of the last time point (**i**) and the survival rates (**j**) are shown. The lower panels of **a–c** were quantifications of colony numbers from three independent experiments presented as mean ± SEM and analyzed by one-way ANOVA. For **d, e, g**, each mouse group contains 20 tumors ($n = 20$). For **h, j**, the Vector control, shRNA1, and shRNA2 groups contain 15, 11, and 10 tumors, respectively ($n = 15, 11$, and 10). For **i**, the Vector control, shRNA1, and shRNA2 group contain 6, 7, and 5 tumors, respectively ($n = 6, 7$, and 5). For **e, i**, the boundary closest to the zero indicates the 25th percentile, a line within the box means the median, and the boundary of the box farthest from zero marks the 75th percentile. Whiskers (error bars) above and below the box indicate the minima and maxima. **d, e, h, i** were presented as mean ± SEM and analyzed by two-sided Student's *t* test. **g, j** was analyzed by two-sided Log-rank test. "*" and "***" denote $P < 0.05$ and $P < 0.001$, respectively. Scale bars: 200 μM (**a–c**). Source data are provided in Source data file.

a poor prognosis in 6 of these 10 types of cancer (Supplementary Fig. 7d). The fact that a low mRNA expression level of CASTOR1 and a high mRNA level of RNF167 predict a poor prognosis of these cancer types suggest the existence of an additional mechanism(s) regulating their mRNA expression. Pharmacological intervention of RNF167 leading to CASTOR1 activation could be considered as a potential therapeutic approach for these cancer types.

CASTOR1 is tumor suppressive in KSHV-induced cellular transformation and lung adenocarcinoma[13,30]. In breast cancer cell lines, the protein level of CASTOR1 appears to be inversely correlated with the level of AKT activation (Supplementary Fig. 8a). Overexpression of CASTOR1 decreases cell proliferation and colony formation in softagar of breast cancer cells while genetic silencing of CASTOR1 has the opposite effect (Fig. 5a–c and Supplementary Fig. 10d–f). In a mouse tumor model, overexpression of WT CASTOR1 inhibits tumor growth and extends animal survival rate (Fig. 5d–g). While the constitutively phosphorylated mutant S14D has a reduced inhibitory effect, the dead phosphorylated mutant inhibits tumor growth even more effective than the WT CASTOR1 (Fig. 5d–g), possibly due to its dominant-negative effect. Hence, our results have demonstrated a tumor-suppressive function of CASTOR1 in breast cancer cells, which is negated by AKT-mediated phosphorylation. Whether CASTOR1 protein has a tumor-suppressive function in other types of cancer remains to be investigated.

In addition to extracellular arginine deficiency commonly observed in tumor microenvironment, the rate-limiting enzyme ASS1 responsible for intracellular de novo arginine synthesis is also frequently silenced in most cancer types[10,11]. These cancer cells are arginine auxotrophic, which are the basis for clinical trials with pegylated arginine deiminase (ADI-PEG20) and human recombinant arginase[10]. These regimens are expected to deprive cancer cells of arginine, leading to CASTOR1 activation, mTORC1 suppression, and tumor regression. While tumors initially respond to ADI-PEG20, ASS1-deficient tumors eventually become resistant to these treatments at least in part by activating the PI3K/AKT pathway[37]. It can be speculated that AKT activation would result in CASTOR1 degradation and mTORC1 activation, contributing to the resistance to the therapies. Hence, AKT-mediated degradation of CASTOR1 could be an important mechanism of resistance to cancer therapies designed to deplete cancer cells of arginine. In this context, combining arginine deprivation and AKT inhibition could be an attractive approach to overcome resistance to these cancer therapies.

## Methods

**Cell culture and transfection**. 293T cells obtained from ATCC (CRL-3216) were maintained in Dulbecco's modified Eagle's medium (DMEM) supplemented with 10% FBS. MCF7, T47D, HCC1569, and HCC202 cells were obtained from Dr. Xiaosong Wang at the University of Pittsburgh and cultured in RPMI1640 with 10% FBS. MM cells were cultured in DMEM plus 10% FBS, while KMM cells were cultured in DMEM plus 10% FBS and 10 µg/ml hygromycin as previously described[26]. HSAEC (FC-0016) and HLBEC (FC-0054) cells were purchased from Lifeline and cultured with the BronchiaLifeTM Epithelial Airway Medium Complete Kit (Lifeline LL-0023). All cells were maintained at 37 C° in 5% CO₂.

For AA deprivation and re-stimulation to assess mTORC1 activation, cells were incubated in Earle's Balanced Salt Solution medium (Thermo 24010043) for 50 min and then stimulated by adding arginine (Sigma A5131) at the indicated concentrations. For ubiquitination assays, cells were deprived of FBS or arginine or treated with 10 µM MK2206 (Selleckchem S1078) overnight before immunoprecipitation and immunoblotting or re-stimulated with FBS or arginine for 12 h before analysis.

For chemical treatments, CHX (CST 2112) or MG132 (Sigma M8699) dissolved in DMSO (VWR 97061-250) was diluted in medium to a specified concentration. Medium containing MG132 or CHX was then used to replace the original medium and cells were cultured in the presence of MG132 for a specified time.

For transfection, Lipofectamine 2000 (Thermo 11668019) was used for transient transfection of plasmids, and RNAimax (Thermo 13778150) was used for transfection of small interfering RNAs based on the manufacturer's instructions.

**Plasmids**. Plasmids purchased from Addgene included: pLKO1-TRC (10878), pcDNA3-myr-HA-AKT1 (46969), pcDNA3-HA-AKT1 (73408), pcDNA3-HA-AKT1-K179M (73409), pcDNA3-HA-AKT1-1-149aa (73410), pcDNA3-HA-AKT1-120-433aa (73411), pRK5-HA-Ubiquitin-WT (17608), pRK5-HA-Ubiquitin-K29 (22903), and pRK5-HA-Ubiquitin-K29R (17602). Plasmids p3.3 empty vector, p3.3-Myc-Ubiquitin-WT, p3.3-Myc-Ubiquitin-K48, p3.3-Myc-Ubiquitin-K63, p3.3-flag-KLHL19, p3.3-flag-KLHL21, p3.3-flag-KLHL22, p3.3-flag-ZNRF1, p3.3-flag-ZNRF2, p3.3-flag-BACURD1, p3.3-flag-BACURD2, p3.3-flag-RNF152, p3.3-flag-RNF167, p3.3-flag-β-Trcp1, p3.3-flag-FBW7, p3.3-flag-HERC5, and p3.3-flag-Skp2 were provided by Jie Chen at Beijing University in China. pcDNA3 empty vector was purchased from Invitrogen. pMD.G and p8.74 were from PlasmidFactory. Human pITA-flag-CASTOR1 WT was cloned from 293T cells. Rat pITA-flag-CASTOR1 WT was previously described[13]. The mutants of human pITA-flag-CASTOR1, including S14A, S14D, K61R, K96R, K213R, K61R/K96R, K61R/K213R, and K61R/K96R/K213R, were generated using a mutagenesis kit (NEB E0554) based on the manufacturer's instructions. The primer sequences used for the cloning are listed in Table S1 and the sequences of all plasmids were confirmed by direct sequencing.

**Antibodies**. Primary antibodies included antibodies to S6K1 (Abcam 32359), pS6K-Thr389 (CST 9205), p4EBP1-Ser65 (CST 9451), 4EBP1 (CST 9644), pan AKT (CST 4691), pAKT-Thr308 (CST 2965), AKT1 (CST 2938), pAKT substrate (RXRXXpS*/T*) (CST 10001), GAPDH (CST 5174), flag (Sigma F1804), flag (Sigma A9594), HA (CST 3724), HA (CST 3444), GST (CST 2625), Ub (Santa Cruz sc-8017), c-Myc (Santa Cruz sc-40), RNF167 (Santa Cruz sc-515405), RNF167 (Proteintech 24618-1-AP), and β-tubulin (Sigma 7B9). Antibodies to CASTOR1 were described as before[13]. Secondary antibodies included mouse anti-Rabbit IgG (Light-Chain Specific) (CST 93702), rabbit anti-Mouse IgG (Light Chain Specific) (CST 58802), goat anti-rabbit horseradish peroxidase (HRP)-conjugated IgG (CST 7074), horse anti-mouse IgG HRP-conjugated IgG (CST 7076), goat anti-mouse IgG DyLight 800 (Bio-Rad STAR117D800GA), and goat anti-rabbit IgG StarBright Blue700 (Bio-Rad 12004161).

**Immunoprecipitation**. Cells were lysed in lysis buffer (50 mM Tris-HCl, with 150 mM NaCl, 1 mM EDTA, and 1% Triton X-100, pH 7.4) supplemented with a complete protease inhibitor cocktail (Thermo 78438) and phosphatase inhibitor (Thermo 78427), followed by centrifugation at 4 °C for 5 min. The supernatant was then precleared with mouse IgG agarose beads (Sigma A0919) at 4 °C for 4 h and subsequently mixed with washed agarose beads conjugated with anti-Flag (Sigma A2220), anti-HA (Thermo 26182), anti-Myc (Sigma A7470, anti-AKT (Cell Signaling Technology 3653), or mouse IgG antibodies (Sigma A0919) at 4 C° overnight. Immunocomplexes were washed extensively 3 times with washing buffer (50 mM Tris-HCl, 150 mM NaCl, pH 7.4). The immunoprecipitates were eluted with 2× sodium dodecyl sulfate (SDS) and then subjected to immunoblotting analysis.

For transfection experiments, $6 \times 10^8$ cells were seeded in 10 cm dishes and transfected with 5 µg of each plasmid using Lipofectamine 2000 (Thermo 11668019) for 48 h. Cells were then treated and lysed as described above.

**Immunoblotting analysis**. To detect all proteins except CASTOR1, samples were separated with 4–20% SDS-polyacrylamide gels (Genscript M00656 and M00657). To detect CASTOR1 protein, samples were resolved with 10% SDS-polyacrylamide gels (Genscript M00665 and M00666). Proteins resolved in gels were then transferred to nitrocellulose membranes (GE Healthcare 10600004), which were incubated with primary and secondary antibodies overnight and for 1 h at room temperature, respectively. The signals were developed using the Luminiata Crescendo Western HRP Substrate (EMD Millipore WBLUR0500) and SuperSignal West Femto Maximum Sensitivity Substrate (Thermo 34096) or fluorescence secondary antibodies. The images were recorded with a ChemiDoc MP Imaging System (Bio-Rad 17001402) at Chemi, Dylight 500, DyLight 800 or StarBright B700 channels.

**In vitro kinase assay**. Recombinant GST-AKT1 protein (Novus Biologicals, 1775-KS) was mixed with GST-CASTOR1 protein (Novus Biologicals, H00652968-P01) in a 30 µl reaction mixture at room temperature for 1 h. The reaction mixture contained protease inhibitors, 100 mM HEPES (pH 7.4), 150 mM NaCl, 50 mM MgCl₂,1 mM dithiothreitol, 0.01% NaN₃, 1 mM ATP, 0.2 µg GST-AKT1, and 1 µg GST-CASTOR1.

**Lentivirus-mediated overexpression and knockdown of genes**. CASTOR1 short hairpin RNAs, non-targeting control (NT), Flag-tagged CASTOR1 WT, S14A, and S14D expression lentiviral plasmids, or the empty vector control pITA was cotransfected with pMDG and p8.74 packaging plasmids into 293T cells using the Lipofectamine 2000 (Thermo 11668019). At day 2 and 3 post-transfection, the supernatant of 293T cells was collected and filtered with a 0.45-µm filter. The

transduction of cells was done by spinning infection at 1500 rpm at room temperature for 1 h with 10 μg/ml polybrene (Sigma A5431). The expression of CASTOR1 was confirmed by immunoblotting at day 3 post-transduction.

**Colony formation in softagar**. A total of $2 \times 10^4$ MCF7 or HCC1569 cells were suspended in 1 ml of 0.3% top agar (Sigma A5431) and then plated onto one well of 0.5% base agar in 6-well plates, which were maintained for 10 or 30 days, respectively. Colonies were photographed with a ×4 objective with an inverted microscope.

**Bromodeoxyuridine (BrdU) incorporation and apoptosis assay**. For BrdU incorporation, MCF7 or HCC1569 cells were pulsed with 10 μM BrdU (Sigma B5002) for 2 h and then fixed with 70% ethanol, permeabilized with 2 M hydrochloric acid, and stained with an anti-BrdU monoclonal antibody (Thermo B35129). Apoptotic cells were detected by co-staining with 4,6-diamidino-2-phenylindole (Sigma D9542) and PE-Cy7 Annexin V Apoptosis Detection Kit (eBioscience 88810374) following the instructions of the manufacturers. Flow cytometry was performed in a BD LSRFortessa system (BD Biosciences) and the analysis was done with FlowJo.

**Reverse transcription real-time quantitative polymerase-chain reaction (RT-qPCR)**. Total RNA was extracted by using TRI Reagent (Sigma T9424) based on the manufacturer's instructions. Total RNA was subjected to RT using the Maxima H Minus First Strand cDNA Synthesis Kit (Thermo K1652). SsoAdvanced™ Universal SYBR® Green Supermix Kit (Bio-Rad 172-5272) was applied for qPCR analysis. The relative mRNA levels were normalized to a house-keeping gene, which yielded $2^{-\Delta\Delta Ct}$ values. For qPCR reaction, each sample was run in triplicates with cycle threshold (Ct) values within 0.5 Ct differences among the triplicates. The primers used for gene expression were 5′GCCACCACCCTCATAGATGT3′ (forward) and 5′AGGAGGTCACTGGGGAACTT3′ (reverse) for human CASTOR1 and ATCATTGCTCCTCCTGAGCG (forward) and CGGACTCGTCATACTCCT GC (reverse) for human β-actin.

**Mouse experiments**. Athymic Nude-Foxn1nu mice were purchased from Envigo. Mice were raised under 12-h light/dark cycle and with standard diet at the University of Pittsburgh. MCF7 cells transduced with a vector control, Flag-CASTOR1 WT, S14A, or S14D were trypsinized and concentrated by centrifugation to $5 \times 10^6$ per 100 μl in DMEM supplemented with 10% FBS. An equal volume of cells was mixed with an equal volume of Matrigel (VWR 47743-720), and then $5 \times 10^6$ cells were subcutaneously injected into each flank of the mouse. The mice were inserted with estrogen pellet (Sigma 8875) before injection. Tumor volume was measured twice a week and calculated based on the formula ($V = L \times W \times W \times 0.5$). Mice were euthanized when the tumor size reached the upper limit of 1500 mm³. All mouse experiments were done following the protocol approved by the University of Pittsburgh Institutional Animal Care and Use Committee (Protocol #: 18073052).

**Quantification, statistical analysis, and reproducibility**. The intensity of a protein band was quantified with the Image Lab Software (Bio-Rad). Data were presented as mean ± SEM (standard error of the mean) and analyzed by two-tailed Student's $t$ test or one-way analysis of variance (ANOVA) if multiple samples were involved followed by Tukey post hoc test if $P < 0.05$. All statistical analyses were done with the Prism software package (PRISM 6.0 and 8.0, GraphPad Software, USA). A $P < 0.05$ was considered as statistically significant. Statistical symbols NS denotes not significant.

**Reporting summary**. Further information on research design is available in the Nature Research Reporting Summary linked to this article.

## Data availability

All data supporting this study are available within this article, the Supplementary file, and the Source data as indicated in the Reporting Summary for this article. Source data are provided with this paper.

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

## Acknowledgements

We would like to thank Dr. Xiaosong Wang for providing the cells and the valuable suggestions of Dr. Patrick S. Moore, Dr. Fred Homa, Dr. Masa Shuda, and Dr. Man-tzu Wang. We also thank S.-J.G.'s laboratory members for technical assistances and helpful discussions. This work was supported by grants from NIH (CA096512, CA124332, CA132637, CA213275, CA177377, DE025465 and CA197153) to S.-J.G.

## Author contributions

S.-J.G. conceived and managed the project. S.-J.G. and T.L. designed the experiments. T.L., X.W., E.J., S.R.d.S., L.C., X.Z., and S.W. performed experiments and analyzed the data. T.L. and S.-J.G. wrote the paper. T.L., S.-J.G., W.S., and E.J. revised the paper.

## Competing interests

The authors declare no competing interests.
