## [Peer Review File · Nature Communications]

Reviewers' comments:

Reviewer #1 (Remarks to the Author):

In the manuscript "RNF167 activates mTORC1 and promotes tumorigenesis by targeting CASTOR1 for ubiquitination and degradation" by Li and colleagues, the authors study the regulation of mTORC1 signaling. CASTOR1 acts as a negative regulator of mTORC1 by sequestering the mTORC1 activator GATOR2. It had previously been found that this role of CASTOR1 could be counteracted by direct binding of arginine, thus linking nutrient availability to mTORC1 activation. The authors now identify the RING E3 ligase RNF167 as yet another regulator of CASTOR1. Specifically, they find that phosphorylation of CASTOR1 Ser14 by Akt promotes the CASTOR1-RNF167 association, followed by CASTOR1 ubiquitylation and degradation. Moreover, RNF167-mediated CASTOR1 degradation activates mTORC1 independent of arginine and promotes breast cancer progression. The finding of this regulatory mechanism is important because it is known that high level of CASTOR1 protein overrides arginine-mediated mTORC1 activation. Furthermore, arginine synthesis is silenced in most types of cancer and extracellular arginine is often limiting in the tumor microenvironment.

The data in general is of very good quality, and the analysis provides evidence in support of the conclusions and the model. Publication in Nature Communications is recommended provided that the following points are addressed.

1. Fig.1a: The evidence that 4EBP1 phosphorylation is reduced is not convincing. Please present a shorter exposure and the quantitation of that exposure.
2. Please discuss the finding that Ser14 phosphorylation seemed to stabilize the binding of Akt to CASTOR1 (e.g., Fig.2a). Has a similar phenomenon been reported for other kinases? This is somewhat counter-intuitive.
3. Figs.3d-e. The important question here is how much free MIOS becomes available in order to activate mTORC1 under these conditions, so a MIOS IP followed by Flag blot would be a more appropriate experimental design.
4. Fig.4b: The effect of S14D is not convincing. I see only one colony that is larger than those in S14A. Please provide quantitation of the data.
5. Methods: in general, methods are not described in sufficient detail for others to be able to reproduce the data. For example, the experiments in Fig.1c are not as trivial as they seem. The authors need to inform how much DNA was used for a certain dish size, how long after transfection were cells lysed, etc.

Minor points

p.5, line 93: Fig1c is a typo. It should be Fig.2c.

p.17, line 328: what does "was manufactured from Invitrogen" mean?

Reviewer #2 (Remarks to the Author):

Previous studies have shown that high levels of CASTOR1 protein override arginine-mediated mTORC1 activation. The authors provide evidence that Arginine withdrawal results in Akt activation, binding and phosphorylation of CASTOR1 at serine14. This promotes its ubiquitination and degradation via the E3 Ub ligase RNF167. RNF167-mediated CASTOR1 degradation activates mTORC1 independent of arginine and promotes breast cancer progression. This is an interesting study that addresses both an additional mechanism of Arginine signaling in mTORC1 regulation as well as the controversy of whether mitogen-dependent Akt signaling contributes only to mTORC1

activation only via TSC2 and raptor phosphorylation or also via amino acids. Additional experimentation is needed to support the authors conclusions as well as a more extensive discussion of the rationale leading to the investigation into Castor1 and importantly, the controversy mentioned above.

Comments and Concerns:

1. In Fig 1a, deprivation of FBS or Arginine leads to a similar reduction in mTORC1 signaling based on p-4EBP1 levels, while changes in Castor1 levels are different. The authors need to address this observation in terms of other nutrients in FBS that could be contributing to differential signaling. Again, FBS has several nutrients, the authors should try to examine specific components such as individual growth factors, LPA, all amino acids, etc. and show that the response is broadly more dependent on FBS as a whole and not on other nutrients.
2. Along these lines, please provide an extensive time course analysis upon removal of arginine to better establish the relationship between mTORC1 inactivation, Akt activation and Castor1 mRNA and protein levels in the absence and presence of FBS. This initial characterization also needs to be completed in more than one cell type.
3. Does chronic Leucine deprivation inactivated mTORC1 signaling, activate Akt and decrease Castor1 protein levels? A similar experiment as requested above would be appropriate.
4. In Figure 2, does FBS removal activate S14 phosphorylation of Castor1 and is that site then sensitive to MKK206. Using the substrate motif antibody of Akt, the authors can check for the phosphorylation of this site in the context of FBS, amino acids, arginine and leucine depletion.
5. In Fig 3a, when comparing +/- Arginine and Castor1 levels, at 0.8 microg of Castor, looks like Arginine does play a role in activation of mTORC1 based on p-4EBP1 levels. This observation needs to be clarified as the authors claim that Castor1 overexpression renders cells insensitive to Arginine dependent mTORC1 activation.
6. In Figure 3d, MIOS levels seem to decrease dependent on Arginine, (Lane 2 vs lane 1). The authors should comment on this, as they claim that when Castor1 is overexpressed, cells are rendered insensitive to Arginine.
7. Is there a physiological system where Castor1 is highly expressed and in that context is Arginine rendered irrelevant? Also, in TSC2-/- cells which should be still sensitive to amino acid depletion, have the authors looked at Castor1 levels and in that context? Is Arginine still important?
8. Previous reports show that Castor depletion specifically renders cells insensitive to Arginine depletion. The authors should address in this article, why they are talking about Castor1 overexpression and if that is physiologically relevant.
9. In Figure 4B, wouldn't the S14D mutant have a much higher tumor growth than even possibly vehicle?

Additional comments.

Genes associated with the 17q23 amplicon are overexpressed in several breast cancer cell lines including T47 and perhaps MCF7 as well. S6K1 is one of these genes and may be involved in negative feedback to Akt/mTOR signaling and also possesses a similar substrate specificity as Akt. The authors should investigate the potential contribution of S6K1 in their findings by using S6K1 inhibitors.

Responses to Reviewers' Comments:

Reviewer #1 (Remarks to the Author):

In the manuscript "RNF167 activates mTORC1 and promotes tumorigenesis by targeting CASTOR1 for ubiquitination and degradation" by Li and colleagues, the authors study the regulation of mTORC1 signaling. CASTOR1 acts as a negative regulator of mTORC1 by sequestering the mTORC1 activator GATOR2. It had previously been found that this role of CASTOR1 could be counteracted by direct binding of arginine, thus linking nutrient availability to mTORC1 activation. The authors now identify the RING E3 ligase RNF167 as yet another regulator of CASTOR1. Specifically, they find that phosphorylation of CASTOR1 Ser14 by Akt promotes the CASTOR1-RNF167 association, followed by CASTOR1 ubiquitylation and degradation. Moreover, RNF167-mediated CASTOR1 degradation activates mTORC1 independent of arginine and promotes breast cancer progression. The finding of this regulatory mechanism is important because it is known that high level of CASTOR1 protein overrides arginine-mediated mTORC1 activation. Furthermore, arginine synthesis is silenced in most types of cancer and extracellular arginine is often limiting in the tumor microenvironment.

The data in general is of very good quality, and the analysis provides evidence in support of the conclusions and the model. Publication in Nature Communications is recommended provided that the following points are addressed.

Response:

We'd like to thank this Reviewer for the positive comments and endorsements of our study.

1. Fig.1a: The evidence that 4EBP1 phosphorylation is reduced is not convincing. Please present a shorter exposure and the quantitation of that exposure.

Response:

Fig. 1a is now Fig. 1c. Based on the suggestion of this Reviewer, we have now given a shorter exposure of p4EBP1 and quantified the relative levels of p4EBP1 normalized to total 4EBP1 levels. The results are presented as Fig. 1c. Furthermore, based on the reviewer's other suggestions (please see later comments), we've now performed kinetic analyses upon deprivation of FBS and arginine in multiple cell lines including 293T, MCF7 and T47D. These new results are added as new panels Fig. 1a, Fig. 1b, and Fig. S1a-j.

2. Please discuss the finding that Ser14 phosphorylation seemed to stabilize

the binding of Akt to CASTOR1 (e.g., Fig. 2a). Has a similar phenomenon been reported for other kinases? This is somewhat counter-intuitive.

Response:

We also think this is an intriguing observation, which has an AKT-mediated feed-forward negative regulatory effect of CASTOR1. Similar phenomenon is observed by Shane R. Horman et. al. entitled “Akt-mediated phosphorylation of argonaute 2 downregulates cleavage and upregulates translational repression of microRNA targets, *Mol Cell* **50**, 356-367 (2013)”. In addition to discussions in the previous submission, we have added more discussions together with this reference (ref# 27) to the revised manuscript (line from line 202-208, 417-420).

3. Figs.3d-e. The important question here is how much free MIOS becomes available in order to activate mTORC1 under these conditions, so a MIOS IP followed by Flag blot would be a more appropriate experimental design.

Response:

This is an excellent suggestion. We have now performed this experiment and added the new results as Fig. 4g, which confirm that the binding affinity to MIOS is significantly lower for S14D than WT and S14A CASTOR1.

4. Fig.4b: The effect of S14D is not convincing. I see only one colony that is larger than those in S14A. Please provide quantitation of the data.

Response:

The images are representative and can only accommodate limited areas and colonies. Based on the suggestion of this Reviewer, we've quantified the number of colonies and added the quantification results for Fig. 5a-c (Fig. 4a-c in previous submission).

5. Methods: in general, methods are not described in sufficient detail for others to be able to reproduce the data. For example, the experiments in Fig. 1c are not as trivial as they seem. The authors need to inform how much DNA was used for a certain dish size, how long after transfection were cells lysed, etc.

Response:

We have now added more detailed descriptions including all essential information related to the experiments in the Methods section.

Minor points

p.5, line 93: Fig1c is a typo. It should be Fig.2c.

Response:

The Reviewer is correct. We have corrected the mistake (please note that previous Fig. 2c is now Fig. 1c).

p.17, line 328: what does “was manufactured from Invitrogen” mean?

Response:

The pCDNA3 was previously purchased from Invitrogen by our lab, but was discontinued now. We have corrected the statement. The map of the vector is attached here for reference.

[REDACTED]

Reviewer #2 (Remarks to the Author):

Previous studies have shown that high levels of CASTOR1 protein override arginine-mediated mTORC1 activation. The authors provide evidence that Arginine withdrawal results in Akt activation, binding and phosphorylation of CASTOR1 at serine14. This promotes its ubiquitination and degradation via the E3 Ub ligase RNF167. RNF167-mediated CASTOR1 degradation activates mTORC1 independent of arginine and promotes breast cancer progression. This is an interesting study that addresses both an additional mechanism of Arginine signaling in mTORC1 regulation as well as the controversy of whether mitogen-dependent Akt signaling contributes only to mTORC1 activation only via TSC2 and raptor phosphorylation or also via amino acids. Additional experimentation is needed to support the authors conclusions as well as a more extensive discussion of the rationale leading to the investigation into Castor1 and importantly, the controversy mentioned above.

Response:

We thank this Reviewer for the endorsements of our study and the suggestions. We've now added new experimental results, which are outlined

below in the responses to the specific comments below and additional discussions of the rationale leading to our study of CASTOR1 in the Introduction (line 46-65).

Comments and Concerns:

1. In Fig 1a, deprivation of FBS or Arginine leads to a similar reduction in mTORC1 signaling based on p-4EBP1 levels, while changes in Castor1 levels are different. The authors need to address this observation in terms of other nutrients in FBS that could be contributing to differential signaling. Again, FBS has several nutrients, the authors should try to examine specific components such as individual growth factors, LPA, all amino acids, etc. and show that the response is broadly more dependent on FBS as a whole and not on other nutrients.

Response:

We completely agree with this Reviewer that FBS contains diverse constituents that can trigger complex cellular signaling pathways. As suggested by this Reviewer, we have now performed additional experiments to examine the roles of different nutrients in FBS including leucine, amino acids and kinetics of deprivation of nutrients in different cell types in regulating CASTOR1, AKT and mTORC1 (Fig. 1a, 1b, 2a, 3b-h and S1k-l). These results show that the CASTOR1 level is negatively correlated with AKT activation including the condition of FBS or arginine deprivation regardless the relationship of AKT and mTORC1 (Fig. 1a-c). Furthermore, AKT inhibitor mimics the effect of FBS deprivation on both upregulating CASTOR1 and inactivating mTORC1 in multiple cells including 293T, T47D and MCF7 (Fig. 1c and S8d), confirming that AKT signaling plays a fundamental role in regulating CASTOR1 protein level regardless of its upstream signals. Hence, the effect of FBS on CASTOR1 is entirely dependent on AKT signaling regardless the specific types of nutrients and signaling. Furthermore, we have now performed knockdown of TSC2 and the results show that AKT- and RNF-167-mediated activation of mTORC1 is independent of TSC2 (Fig. 4d), and that RNF-167 knockdown alone in cells with high CASTOR1 expression is sufficient to robustly activate mTORC1 (Fig. 3g). In addition to adding the new results, we have also added discussions in the relative sections in Results and Discussion.

2. Along these lines, please provide an extensive time course analysis upon removal of arginine to better establish the relationship between mTORC1 inactivation, Akt activation and Castor1 mRNA and protein levels in the absence and presence of FBS. This initial characterization also needs to be completed in more than one cell type.

Response:

Based on the suggestions of this Reviewer, we've performed a time-course deprivation of either arginine or FBS, and examined AKT activation, mTORC1 activity and CASTOR1 expression at protein and mRNA levels in 293T, MCF7 and T47D cells. The data is added into the revised manuscript and shown as Fig. 1a, 1b, and S1a-j. The results show that the level of CASTOR1 depends on the AKT status.

3. Does chronic Leucine deprivation inactivated mTORC1 signaling, activate Akt and decrease Castor1 protein levels? A similar experiment as requested above would be appropriate.

Response:

Based on the suggestions of this Reviewer, we've performed the chronic deprivation of leucine followed by examining AKT activation, mTORC1 activity, CASTOR1 protein and mRNA levels. Similar to arginine deprivation, leucine deprivation inactivated mTORC1, which caused AKT activation and hence decreased CASTOR1 expression at both protein and mRNA levels.

These new results are added into the revised manuscript and shown as Fig. S1k, l.

4. In Figure 2, does FBS removal activate S14 phosphorylation of Castor1 and is that site then sensitive to MKK206. Using the substrate motif antibody of Akt, the authors can check for the phosphorylation of this site in the context of FBS, amino acids, arginine and leucine depletion.

Response:

Our new and previous results demonstrate that FBS removal and MK2206 inactivates whereas arginine deprivation activates AKT (new Fig. 1a, c and S1c, e). Specifically, AKT inhibitor MK2206 reduces CASTOR1 phosphorylation at S14 and increased CASTOR1 protein level by inactivating AKT (Fig. 1c, 2c, 2d and Fig. S8d). As suggested by this Reviewer, we've also performed experiments with deprivation of total amino acids, FBS and arginine. FBS and total amino acids deprivation decreased whereas arginine deprivation alone increased CASTOR1 phosphorylation at S14, which is positively correlated to AKT activation. Intriguingly, we didn't see much change of CASTOR1 phosphorylation at S14 following leucine deprivation, which might be explained by its relatively less AKT activation than arginine deprivation. These new results are added as Fig. 2a in the revised manuscript.

5. In Fig 3a, when comparing +/- Arginine and Castor1 levels, at 0.8 microg of Castor, looks like Arginine does play a role in activation of mTORC1 based on p-4EBP1 levels. This observation needs to be clarified as the authors claim that Castor1 overexpression renders cells insensitive to Arginine dependent mTORC1 activation.

Response:

We completely agree with the Reviewer. Indeed, when >0.4 microgram of CASTOR1 DNA was used, the cells didn't respond to arginine. It needs to be pointed out that we observed marginally increase of p4EBP1 but no noticeable change of pS6K after overexpressing 0.8 µg CASTOR1 before and after arginine stimulation. It has been previously reported that amino acids mediated phosphorylation of 4EBP1 is primarily dependent on leucine rather than other amino acids (Fox et al., Amino acid effects on translational repressor 4E-BP1 are mediated primarily by L-leucine in isolated adipocytes. Am J Physiol. 1998, 275:C1232-8). This is consistent with our observation that p4EBP1 is less regulated by arginine and CASTOR1 than pS6K (Fig. 1b, 4c, e, and Fig. S1e, f and I). Therefore, pS6K is more representative in this situation.

6. In Figure 3d, MIOS levels seem to decrease dependent on Arginine, (Lane 2 vs lane 1). The authors should comment on this, as they claim that when Castor1 is overexpressed, cells are rendered insensitive to Arginine.

Response:

Based on the previous studies, the CASTOR1 homodimer binds to arginine with a K_m around 34.8 μM , and arginine regulates mTORC1 activity by interrupting CASTOR1 and MIOS interaction in a dose-dependent manner (Chantranupong L, *et al.* The CASTOR Proteins Are Arginine Sensors for the mTORC1 Pathway. *Cell* 165, 153-164 (2016). Therefore, a prerequisite that CASTOR1 completely overrides arginine-mediated mTORC1 activation is that cells must overexpress sufficient amount of CASTOR1 protein to sequester MIOS except the portion that is negated by arginine. This is supported by our results in Fig. 3a and 4a. In Fig. 3d (now is Fig. 4c in the revised manuscript), insufficient CASTOR1 protein is expressed. Hence, MIOS is still partially released and mTORC1 is activated upon arginine stimulation, further supporting the observation that CASTOR1 protein level is critical in regulating mTORC1, which can be negated by AKT1.

7. Is there a physiological system where Castor1 is highly expressed and in that context is Arginine rendered irrelevant? Also, in TSC2^{-/-} cells which should be still sensitive to amino acid depletion, have the authors looked at Castor1 levels and in that context? Is Arginine still important?

Response:

Following the suggestions of this reviewer, we've examined the CASTOR1 protein level in several cell types, and then assessed the mTORC1 activity after arginine deprivation or re-stimulation. We have found that cells expressing low CASTOR1 such as MCF7 cells (Fig. 3b), mTORC1 is responsive to arginine re-stimulation following deprivation (Fig. 3e). In contrast, cells with high endogenous CASTOR1 protein levels such as T47D, HLBEK and HSAEC are not responsive to arginine-mediated mTORC1 activation (Fig. 3b, e-h). Interestingly, cells with almost no detectable CASTOR1 such as Hela cells have high constitutive mTORC1 activity and are resistant to arginine deprivation and re-stimulation. CASTOR1 is likely constitutively inactivated in these cells by an unclear mechanism. However, it is entirely possible that this is through a constitutive AKT- and RNF167-mediated mechanism.

We have also performed knockdown of TSC2 and the results show that the cells indeed remain sensitive to FBS deprivation (Fig. 4d). These new results indicate that AKT- and RNF-167-mediated activation of mTORC1 is independent of TSC2. Furthermore, RNF-167 knockdown alone in cells with high CASTOR1 expression is sufficient to robustly activate mTORC1 (Fig. 3g).

8. Previous reports show that Castor depletion specifically renders cells insensitive to Arginine depletion. The authors should address in this article, why they are talking about Castor1 overexpression and if that is physiologically relevant.

Response:

We completely agree with the Reviewer that CASTOR1 depletion will render cells insensitive to arginine deficiency and constitutively activate mTORC1 signaling pathway, which is indeed observed in Hela cells (Fig 3c, d). In contrast, other cells including T47D, HLBEc, HSAEC with high endogenous CASTOR1 protein level that might mimic the condition of CASTOR1 overexpression in 293T cells override arginine-mediated mTORC1 activation in physiological conditions (Fig 3b, e, f, h). Furthermore, RNF-167 knockdown alone in cells with high CASTOR1 expression is sufficient to robustly activate mTORC1 (Fig. 3g). These results indicate that AKT- and RNF167-mediated activation of mTORC1 through downregulation of CASTOR1 is functional in physiological conditions. Our results (Fig. 3) reveal that the response of cells to arginine is controlled by endogenous CASTOR1 protein level, which further demonstrates the importance of fine-tuning of CASTOR1 protein level in regulating mTORC1 and cell proliferation.

9. In Figure 4B, wouldn't the S14D mutant have a much higher tumor growth than even possibly vehicle?

Response:

We suppose that the Reviewer asks whether S14A has a dominant negative effect? Indeed, we have observed that the tumor suppressive effect S14A is slightly stronger than WT CASTOR1 (Fig. 5d-g). Since the effect is marginal albeit significant, the results need to be further confirmed.

Additional comments.

Genes associated with the 17q23 amplicon are overexpressed in several breast cancer cell lines including T47 and perhaps MCF7 as well. S6K1 is one of these genes and may be involved in negative feedback to Akt/mTOR signaling and also possesses a similar substrate specificity as Akt. The authors should investigate the potential contribution of S6K1 in their findings by using S6K1 inhibitors.

Response:

As suggested by this Reviewer, we've examined the effect of a S6K1 specific inhibitor on CASTOR1 in 293T cells. The S6K1 inhibitor doesn't activate AKT, and hence has no effect on the CASTOR1 protein level. We have added the new results in Fig. S1I.

Reviewers' comments:

Reviewer #1 (Remarks to the Author):

The authors have addressed all criticism in a positive and satisfactory manner, and with that generating an improved manuscript. I have no further comments.

Reviewer #2 (Remarks to the Author):

I remain concerned about the lack of correlation between Akt phosphorylation and Castor1 protein levels, especially in T47D and MCF7 cells. For example, in the new time course experiments presented, pAkt is high before starvation and nearly completely disappears by 15'-30' and do not change much until 8-24h when Akt is being re-phosphorylated. Furthermore, at these later time points, when Castor1 is increasing, pS6K remains largely unchanged. Another example is in new figure s11 where again there is a lack of correlation between increased pAkt upon Arg and Leu starvation, S6K and 4EBP1 phosphorylation, and S6K and 4EBP1 mobility shifts which reflect phosphorylation. For example, upon Leu starvation the authors show decreased pS6k but a reduced mobility in SDS-PAGE which is consistent with its increased phosphorylation and similarly, you show decreased p4EBP1 but the mobility shift suggests hyperphosphorylation. Since the phosphorylation sites for the phospho-specific antibodies are not described (or at least not easy to find), this also makes it difficult to evaluate the data since phosphorylation at different sites reflects differences in activation state and signaling. Similarly, in new figure 2a how do the reviewers account for Leu starvation-dependent activation of Akt (pAkt) without an increase in Castor1 phosphorylation (RXRXXpS/T)? Also, in fig. 2d how does the allosteric Akt inhibitor, MK2206, reduce the pAkt signal in vitro ?

In the response to comment 5 the authors cite Fox et al. to support their argument but this study is not consistent with work in other cell systems.

Reviewer #2 (Remarks to the Author):

I remain concerned about the lack of correlation between Akt phosphorylation and Castor1 protein levels, especially in T47D and MCF7 cells. For example, in the new time course experiments presented, pAkt is high before starvation and nearly completely disappears by 15'-30' and do not change much until 8-24h when Akt is being re-phosphorylated.

Response: We respectfully disagree with this Reviewer's assessment of these results. Unlike the results of knockdown or overexpression experiments, which we expect an inverse correlation of AKT phosphorylation with CASTOR1 protein level at a static time point, these are kinetic experiments, which we expect that the change of pAKT would precede that of CASTOR1 level. This is exactly what we have observed following FBS deprivation and arginine deprivation in both T47D and MCF7 cells. As you can see from the figure panels below that FBS deprivation in MCF7 and T47D cells inactivated AKT at as early as 15 min and remained low. CASTOR1 protein level indeed started to increase at 8 h. This is consistent with our observation that CASTOR1 protein degradation took about 8 h (Fig. S5b). For arginine deprivation, there was a moderate fluctuation of pAKT from 15 to 60 min. pAKT level was slight reduced at 2 and 4 h following arginine deprivation in MCF7 and T47D cells, which led to a moderate increase of CASTOR1 protein around 8 and 16 h. However, pAKT was significantly increased at 16 h, which was caused by the well-known feedback mechanism of mTORC1 inhibition. We can also clearly see CASTOR1 protein level decreased after at 24 h. All these results clearly showed that CASTOR1 protein levels were dependent on AKT phosphorylation status with the decrease of CASTOR1 protein level closely trailing behind AKT activation in both MCF7 and T47D cells.

We apologize that we did not make this clear in the previous submission but have added additional discussions regarding these results in the revised manuscript (line 115-152).

Furthermore, at these later time points, when Castor1 is increasing, pS6K remains largely unchanged.

Response: It is well-known that the mechanism mediating mTORC1 and its downstream targets pS6K and p4EBP is complex, which requires both nutrients and growth factors. This is a well-studied phenomenon that has been reported by many labs. At the later time points following FBS starvation (24 h), the pS6K and p4EBP levels were not further decreased despite the increased CASTOR1 level. This was due to the already low mTORC1 activity, and already low pS6K and p4EBP levels because of the lack of growth factors (e.g. FBS). As a result, the pS6K and p4EBP levels could not be further decreased. Additionally, in MCF7 and T47D cells, pAKT was weakly activated at later time points due to the compensatory feedback of mTORC1 inhibition, which is well-known and has been reported by many different labs. This feedback pAKT activation also maintained pS6K and p4EBP levels despite the increased CASTOR1 level. Hence, at the later time points, mTORC1 or its downstream effectors pS6K and p4EBP were concomitantly regulated by both positive regulator AKT and negative regulator CASTOR1, and therefore cannot be only assessed by the CASTOR1 protein level alone.

Similarly, following arginine starvation, the pS6K and p4EBP levels were not further decreased at the later time points despite the increased CASTOR1 level because the

pS6K and p4EBP levels were already at extremely low levels. Consistent with these results, the decrease of CASTOR1 level at 24 h led to a slight increase of pS6K and p4EBP levels.

We have added additional discussions regarding these results in the revised manuscript (line 115-152).

Another example is in new figure s1I where again there is a lack of correlation between increased pAkt upon Arg and Leu starvation, S6K and 4EBP1 phosphorylation, and S6K and 4EBP1 mobility shifts which reflect phosphorylation. For example, upon Leu starvation the authors show decreased pS6k but a reduced mobility in SDS-PAGE which is consistent with its increased phosphorylation and similarly, you show decreased p4EBP1 but the mobility shift suggests hyperphosphorylation. Since the phosphorylation sites for the phospho-specific antibodies are not described (or at least not easy to find), this also makes it difficult to evaluate the data since phosphorylation at different sites reflects differences in activation state and signaling.

Response: As described in our response to the previous comments, the activation of mTORC1 and its downstream targets S6K and 4EBP1 is complex, and requires both growth factors (FBS) and nutrients (amino acids). Both leucine and arginine are required for mTORC1 activation. The observed higher levels of pAKT following leucine or arginine starvation was due to the mTORC1 feedback mechanism, which was not expected to lead to activation of downstream targets S6K and 4EBP1 because of the lack of leucine or arginine. Our results are consistent with these well-known facts. Please note that the increase of pAKT level indeed led to the decrease of CASTOR1 protein levels following deprivation of either leucine or arginine (Fig. 1c and Supplementary Fig. S1I).

For the detection of phosphorylation of S6K and 4EBP1, the best approach is to use specific phosphorylation antibodies rather than depending on mobility shift in a SDS-PAGE gel because of the subtle change of the mobility shift is not always reliable and reproducible. This is the reason that all the phosphorylation antibodies are developed. Arginine or leucine deprivation can cause the change of phosphorylation of one amino acid of S6K or 4EBP1, which only results in subtle the mobility shifts on SDS-PAGE. In fact, no obvious shift of S6K or 4EBP1 was observed in our results following either FBS, arginine or leucine deprivation (Fig. 1a-c, Fig. 2a, Fig. 3a-g, Fig. 4b and Supplementary Fig. S1). These results are consistent with those reported in many other published articles (please see references: Chen et al., 2018, KLHL22 activates amino-acid-dependent mTORC1 signalling to promote tumorigenesis and ageing, *Nature*, **557**:585–589; Chantranupong, et al., 2016, The CASTOR proteins are arginine sensor for the mTORC1 pathway, *Cell*, 165:153-164). Furthermore, there are other post-translational modifications such as ubiquitination, which can also affect protein mobility. Hence, phosphorylation specific antibodies are

the best tools to detect these modifications. For our study, we used specific antibodies for pS6K-Thr389 (CST 9205), p4EBP1-Ser65 (CST 9451) and pAKT-Thr308 (CST 2965) for our experiments, which can be found in methods antibody section.

Similarly, in new figure 2a how do the reviewers account for Leu starvation-dependent activation of Akt (pAkt) without an increase in Castor1 phosphorylation (RXRXXpS/T)?

Response: Our results showed that AKT activation was slightly weaker following leucine deprivation than arginine deprivation (Fig. 2a and Fig. S1I), which also led to a slightly weaker CASTOR1 phosphorylation (RXRXXpS/T). We've rerun these experiments and obtained similar but better correlation results (Figure below). We have replaced the figure panel in Fig 2a to better illustrate our conclusion in the revised manuscript. These results clearly showed the positive correlation of pAKT level with CASTOR phosphorylation level and the negative correlation of pAKT level with CASTOR protein level.

Also, in fig. 2d how does the allosteric Akt inhibitor, MK2206, reduce the pAkt signal in vitro?

Response: MK2206 inhibits the activity of AKT in a non-ATP competitive manner, and further results in the inhibition of the PI3K/Akt signaling pathway, cell proliferation and the induction of cell apoptosis. AKT possesses a protein domain called Pleckstrin Homology (PH) domain, which binds with high affinity to phosphoinositides including PIP3 and PIP2. Although the mechanism of action of MK-2206 is not entirely understood, existing data suggest that binding of the inhibitor induces a closed conformation change that occludes the binding sites for the activating kinases PDK1

and mTORC2, and AKT substrates (please see reference: Craig Cherrin, et al., An allosteric Akt inhibitor effectively blocks Akt signaling and tumor growth with only transient effects on glucose and insulin levels in vivo. *Cancer Biol Ther.* 2010, 9:493-503). However, how does MK2206 decrease pAKT level in vitro is not entirely clear. We have repeated the experiment and now used two independent pAKT antibodies, which detected pAKT at S473 and T308, respectively (Figure below). Our new results confirmed our previous results that MK2206 inhibited pAKT at T308. Interestingly, there was no effect on S473. The inhibition of T308 was sufficient to abolish CASTOR1 phosphorylation (p-RXRXXS/T), suggesting that AKT phosphorylation at T308 was required for its phosphorylation of CASTOR1.

In the response to comment 5 the authors cite Fox et al. to support their argument but this study is not consistent with work in other cell systems.

Response: We agree with this reviewer that leucine deprivation is not relevant here as the experiment was performed with and without arginine. Hence the article does not fully support the argument. Nevertheless, our results clearly showed that high doses of CASTOR1 rendered both S6K and 4EBP insensitive to arginine. This is stated in the manuscript (line 299-300).

REVIEWERS' COMMENTS

Reviewer #2 (Remarks to the Author):

The authors have satisfactorily addressed my major concerns.

Response to Reviewer's Comments

Reviewer #2 (Remarks to the Author):

The authors have satisfactorily addressed my major concerns.

Response: We thank the reviewer for supporting our work.